# Multiscale chemogenetic dissection of fronto-temporal top-down regulation for object memory in primates

Toshiyuki Hirabayashi [1] ✉, Yuji Nagai [1], Yuki Hori[1], Yukiko Hori [1], Kei Oyama[1], Koki Mimura[1], Naohisa Miyakawa[1], Haruhiko Iwaoki[1], Ken-ichi Inoue [2], Tetsuya Suhara[1], Masahiko Takada [2], Makoto Higuchi [1] & Takafumi Minamimoto [1]

Visual object memory is a fundamental element of various cognitive abilities, and the underlying neural mechanisms have been extensively examined especially in the anterior temporal cortex of primates. However, both macroscopic large-scale functional network in which this region is embedded and microscopic neuron-level dynamics of top-down regulation it receives for object memory remains elusive. Here, we identified the orbitofrontal node as a critical partner of the anterior temporal node for object memory by combining whole-brain functional imaging during rest and a short-term object memory task in male macaques. Focal chemogenetic silencing of the identified orbitofrontal node downregulated both the local orbitofrontal and remote anterior temporal nodes during the task, in association with deteriorated mnemonic, but not perceptual, performance. Furthermore, imaging-guided neuronal recordings in the same monkeys during the same task causally revealed that orbitofrontal top-down modulation enhanced stimulus-selective mnemonic signal in individual anterior temporal neurons while leaving bottom-up perceptual signal unchanged. Furthermore, similar activity difference was also observed between correct and mnemonic error trials before silencing, suggesting its behavioral relevance. These multifaceted but convergent results provide a multiscale causal understanding of dynamic top-down regulation of the anterior temporal cortex along the ventral fronto-temporal network underpinning short-term object memory in primates.

Visual object memory is a fundamental element that underpins our various cognitive abilities; its underlying critical cortical areas[1–7] and local neuronal mechanisms[8–11] have long been examined especially in primates. Akin to many other cognitive functions, visual object memory is thought to be implemented by brain-wide network of distributed regions and their dynamic interactions[12–19]. Specifically, the anterior ventral temporal cortex (aVTC), in which neurons represent complex visual objects as the final stage of the ventral visual pathway[20–22], has been established to be a crucial node for visual object memory in both humans and non-human primates[1–3,5,6]. For object memory, relevant object representations in the aVTC are thought to be specifically activated with the aid of top-down regulation without bottom-up visual input[13,14,17,19]. However, the brain-wide functional network in which the aVTC is embedded and how the aVTC is top-down regulated

[1]Advanced Neuroimaging Center, National Institutes for Quantum Science and Technology, Chiba 263-8555, Japan. [2]Center for the Evolutionary Origins of Human Behavior, Kyoto University, Inuyama, Aichi 484-8506, Japan. ✉e-mail: hirabayashi.toshiyuki@qst.go.jp

for object memory remains elusive. Local perturbation of a network node identified via unbiased whole-brain functional imaging and subsequent simultaneous perturbation and imaging would enable both causal and data-driven understanding of such hitherto unknown, large-scale network operation[23–28].

Beyond a macroscopic view of network operations, imaging-guided neuronal recordings[29,30] at a node exhibiting perturbation-induced remote activity change[31] would provide a causal understanding of the specific content and dynamics of information flow along the network at a microscopic cellular level. Specifically, although the delay period activity in a memory task retaining the identity of the to-be-remembered object without visual input has been recorded extensively from the aVTC neurons[8,9,11,32], its source and underlying network mechanisms have not as yet been causally identified. Therefore, both the macroscale identification of network nodes and subsequent microscale, cellular-level understanding of causal information flow along the identified nodes are required for a comprehensive understanding of the network mechanisms underpinning object memory.

The recently developed chemogenetic approach enables repetitive and minimally invasive perturbation of virtually the same neuronal population in a brain region of any size, and is thus ideal for causally investigating functions of large primate brain[27,33–40]. Taking advantage of the low invasiveness and high reproducibility, chemogenetics may further provide an invaluable opportunity for multi-step network investigation; neuroimaging and subsequent electrophysiology concurrent with the same perturbation can be performed in the same individual during the same task. Here, by combining chemogenetics with whole-brain functional imaging and neuronal recordings in behaving macaques, we causally investigated the functions of the fronto-temporal network underpinning object memory at both macro- and micro-scales in the same individuals during the same task.

We first identified the ventral fronto-temporal network for the retention of object memory in a data-driven manner by combining functional [$^{15}$O]H$_2$O-positoron emission tomography (fPET)[41,42] during a delayed matching-to-sample (DMS) task with resting-state functional magnetic resonance imaging (rs-fMRI) in macaques. Next, we virally transduced inhibitory designer receptors exclusively activated by designer drug (inhibitory DREADD, hM4Di)[43] for neuronal silencing into the orbitofrontal node, which specifically exhibited both prominent mnemonic activity and functional connectivity with the aVTC node (we address the issue of the possible involvement of other brain regions, such as the lateral prefrontal cortex (LPFC), in the "Discussion"). We then causally examined whether the focal silencing of the orbitofrontal node affected the top-down regulation of the mnemonic activity in the aVTC macroscopically in association with declined mnemonic performance. Finally, we microscopically investigated how were the mnemonic representation of the to-be-remembered object in individual aVTC neurons of the same macaques top-down regulated by the OFC node using the same perturbation during the same task and whether the degree of the resultant remote activity modulation of the aVTC neurons during the delay period in a given trial predicted the mnemonic performance of the trial.

## Results

### Identifying the ventral fronto-temporal network for object memory by combining task fPET with rs-fMRI connectivity

Two macaque monkeys performed a DMS task with visual objects under short (0.3 s) and long (4 and 3 s for monkeys 1 and 2, respectively) delay conditions in a PET scanner (Fig. 1a left, b and Supplementary Fig. 1a). In this task, monkeys were required to retain the cued object memory during the delay period without visual input. Both cue and distractor stimuli were chosen from the same pool of familiar objects. Mnemonic brain activity was extracted as the difference in the regional cerebral blood flow (rCBF) between the short- and long-delay trials. In addition to the expected activation of the aVTC ($p < 0.002$),

prominent mnemonic activity was observed in the orbitofrontal cortex (OFC) ($p < 0.001$) (Fig. 1c) consistently in both monkeys (Supplementary Fig. 1b). The anterior cingulate cortex (ACC) and the intraparietal sulcus (IPS) also exhibited significant mnemonic activity ($p < 0.001$) (Supplementary Fig. 1c); in contrast, the lateral prefrontal cortex (LPFC) was relatively absent of the mnemonic activity in the current paradigm (Supplementary Fig. 1d) (see "Discussion").

To identify the network in which aVTC is embedded, we examined the rs-fMRI connectivity calculated using a dataset obtained from a separate population of twenty macaques[44], taken from the PRIME-DE open database[45], with the fPET-identified aVTC node as a seed region. We found that the aVTC node exhibited significant functional connectivity with the OFC node ($p < 0.002$) (Fig. 1d, e), consistent with previous anatomical studies[46,47]. In contrast, the functional connectivity with the aVTC node was not observed for ACC/IPS nodes ($p > 0.9$) (Fig. 1e and Supplementary Fig. 1e) or LPFC region (Supplementary Fig. 1e). These results suggest that the identified aVTC node, together with the OFC node and its surrounding region, constitutes the ventral fronto-temporal network for object memory[17,48].

### Mnemonic behavioral impact of fPET-guided chemogenetic OFC silencing

To causally investigate the top-down regulation of the aVTC node along the identified ventral fronto-temporal network and its impact on mnemonic performance, we genetically induced hM4Di into the fPET-identified OFC node of the same individuals via adeno-associated viral (AAV) vector injection, which enabled neuron-specific expression of inhibitory DREADD (Fig. 1a)[35,49]. Before silencing experiments, hM4Di expression was confirmed in vivo with [$^{11}$C]deschlorocozapine (DCZ)-PET[27,35,38,39] (Fig. 1f, g and Supplementary Fig. 1f), which was further validated with postmortem immunohistochemistry (Fig. 1h).

Behavioral tests with a DMS task revealed that DREADD agonist administration significantly impaired the mnemonic performance in individual hM4Di-expressing macaques [6- and 9-s delay, $p < 0.02$, unpaired $t$-test; DCZ and Clozapine N-oxide (CNO) were used for monkeys 1 and 2, respectively] (Fig. 2a and Supplementary Fig. 2a), demonstrating that the fPET-identified OFC activity exerted a causal influence on mnemonic performance beyond mere correlation with the mnemonic load. Critically, the OFC silencing-induced deficit was more severe in longer delay conditions [two-way analysis of variance (ANOVA), interaction between factors of OFC silencing and delay length, $p < 0.005$] (Fig. 2a and Supplementary Fig. 2a), suggesting that the OFC activity is more essential when monkeys have to retain the object memory for longer time. In contrast, OFC silencing did not affect the performance in the control condition with the shortest (0.3 s) delay period (Fig. 2a and Supplementary Fig. 2a), suggesting that the observed deficit was not due to impairment in perception, decision-making, rule-based behavior, and/or motor function. Monkeys sometimes made an inappropriate lever release before the choice period, especially in the longest (9 s) delay condition. These refusals, a behavioral signature of reduced motivation for task engagement, did not increase by OFC silencing ($p > 0.5$, unpaired $t$-test) and their frequency did not correlate with the task performance during OFC silencing ($p > 0.4$). We also confirmed that two different DREADD agonists [Clozapine N-oxide (CNO) and DCZ] induced similar behavioral changes (Fig. 2a right and Supplementary Fig. 2b) for CNO and DCZ in the same monkey, respectively. Three-way ANOVA showed that there was no significant interaction across factors of Agonist, Silencing, and Delay length ($p > 0.1$)], indicating that the observed behavioral impact of OFC silencing cannot be explained by peripheral properties of a specific DREADD agonist. Critically, agonist administration before DREADD transduction did not affect mnemonic performance ($p > 0.8$) (Fig. 2b), confirming that the DREADD agonist affected mnemonic performance via hM4Di as intended, rather than via unwanted off-target actions.

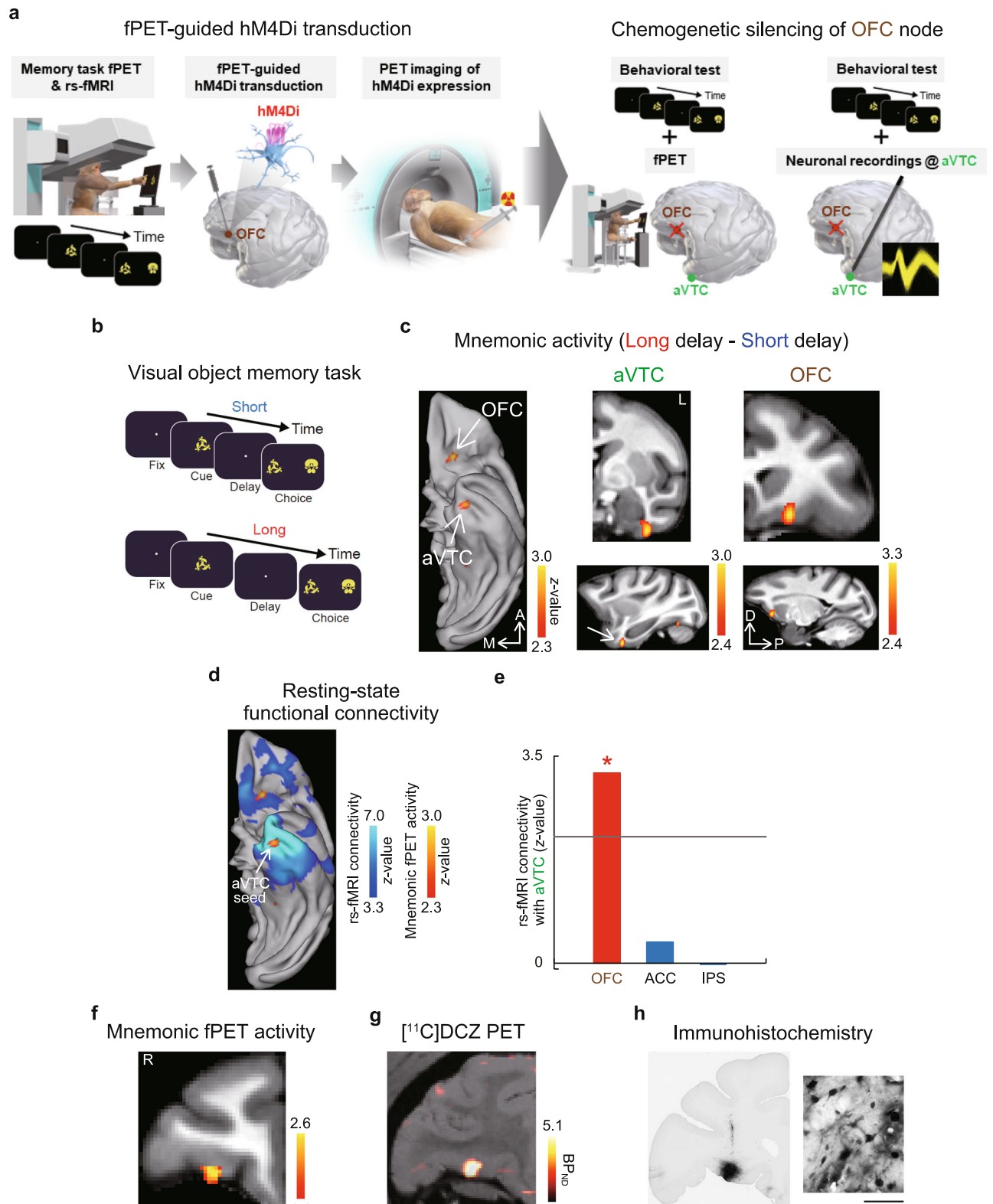

**a** fPET-guided hM4Di transduction — Chemogenetic silencing of OFC node

**b** Visual object memory task

**c** Mnemonic activity (Long delay - Short delay) — aVTC / OFC

**d** Resting-state functional connectivity

**e**

**f** Mnemonic fPET activity

**g** [¹¹C]DCZ PET

**h** Immunohistochemistry

## Chemogenetic fPET during a DMS task reveals brain-wide network effects of focal OFC silencing

To causally dissect the network dysfunction underlying the aforementioned memory-specific behavioral impairment, we next examined the effect of the focal OFC silencing on the whole-brain mnemonic activity. In the vehicle-treated control condition, significant mnemonic activity was reproduced at both the OFC ($p < 0.002$) and aVTC ($p < 0.001$) along the ventral fronto-temporal network following

hM4Di expression at the OFC node (Fig. 3a top, 3b left blue and Supplementary Fig. 3a top; Supplementary Fig. 3b blue for ROI analysis in Fig. 3b blue in each monkey; Supplementary Fig. 3c top for whole-brain analysis in each monkey; Supplementary Fig. 3d, e for rs-fMRI connectivity seeded from activation sites in the OFC and aVTC in each monkey before and after DREADD expression). Following DREADD agonist administration, the mnemonic OFC activity was significantly attenuated ($p < 0.001$, vehicle vs. DREADD agonist, unpaired $t$-test)

**Fig. 1 | Whole-brain fPET mapping of mnemonic activity and fPET-guided DREADD transduction into the OFC node. a** Experimental design. Task fPET-guided transduction of inhibitory DREADD into the prefrontal activation site was followed by simultaneous silencing and fPET or electrophysiology during the same task for multi-scale causal exploration of the fronto-temporal top-down regulation for object memory. **b** DMS task with short (top) or long (bottom) delay period. **c** Mnemonic fPET activity in the ventral fronto-temporal network of two monkeys. $n = 64$ and 68 scans for long and short delay conditions, respectively. Left, ventral view of the macaque brain. L left. **d** Mnemonic fPET activity overlaid on the rs-fMRI connectivity map with the fPET-identified aVTC activation site as a seed region. **e** Statistical test for the strength of rs-fMRI connectivity with the aVTC activation

site as a seed region. Horizontal gray line, threshold for statistical significance ($p = 0.05$, Bonferroni-corrected for multiple comparisons). *$p = 0.00195$, test of no correlation, one-tailed, Bonferroni-corrected. Source data are provided as a Source Data file. **f–h** fPET-guided transduction of inhibitory DREADD into the OFC node. **f** Mnemonic OFC activity in monkey 1. R right. $n = 34$ and 39 scans for long and short delay conditions, respectively. **g** Expression of hM4Di at the OFC activation site visualized with [¹¹C]DCZ-PET in monkey 1. $BP_{ND}$, binding potential. **h** A representative image of histologically verified green fluorescent protein (GFP) expression co-transduced with hM4Di. Scale bars, 3 mm (left) and 50 μm (right). The immunostaining was independently repeated twice.

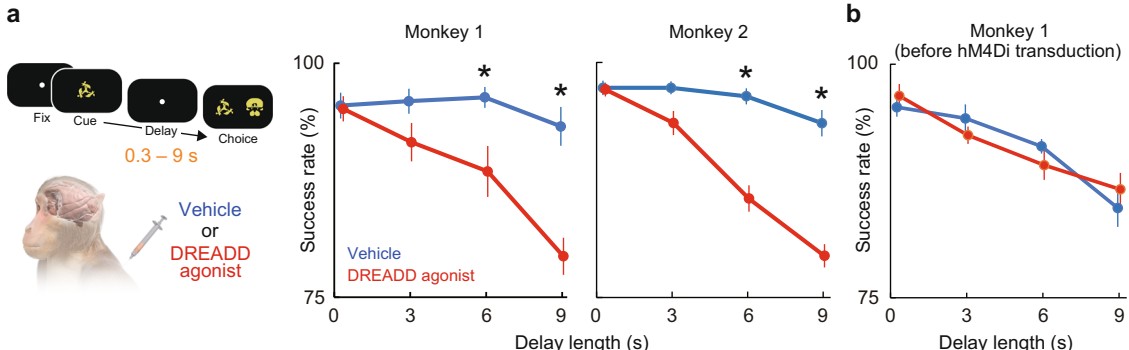

**Fig. 2 | Mnemonic behavioral impact of fPET-guided chemogenetic OFC silencing. a** Behavioral impact of OFC silencing in a DMS task. *: $p = 0.017$ and $0.00027$ for 6- and 9-s delay in monkey 1, $1.1 \times 10^{-5}$ and $2.3 \times 10^{-6}$ for 6- and 9-s delay in monkey 2, two-sided paired $t$-test following two-way ANOVA with interaction between factors of delay length and DREADD agonist administration ($p = 0.0041$ and $1.3 \times 10^{-8}$ for monkeys 1 and 2, respectively). $n = 8$ (8) and 10 (10) sessions for

vehicle (DREADD agonist) condition in monkeys 1 and 2, respectively. DCZ and CNO were used for monkeys 1 and 2, respectively. **b** Behavioral effect of DREADD agonist before hM4Di transduction in monkey 1. Two-way ANOVA, no significant interaction ($p = 0.76$ and $0.49$ for 6- and 9-s delay, $n = 7$ and 5 sessions for vehicle and DREADD agonist condition, respectively). For (**a**, **b**), error bars are sem, and source data are provided as a Source Data file.

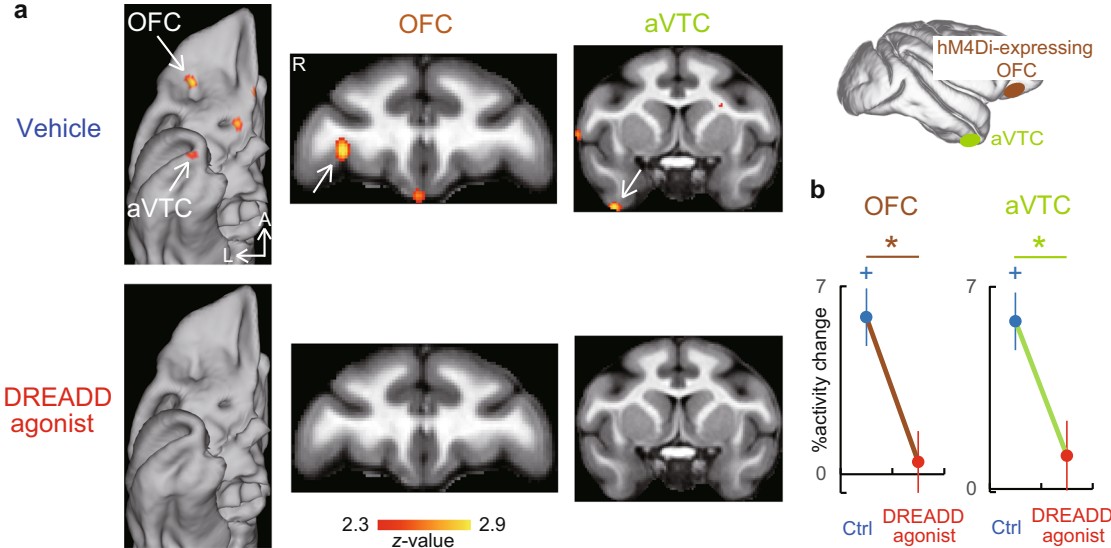

**Fig. 3 | Chemogenetic fPET reveals the impact of OFC silencing on mnemonic activity along the ventral fronto-temporal network. a** Mnemonic fPET activity along the ventral fronto-temporal network under vehicle (top, $n = 130$ and 128 scans for long- and short-delay conditions, respectively) and DREADD agonist (bottom, $n = 100$ and 94 scans for long- and short-delay conditions, respectively) administration conditions in the two monkeys. A anterior, L lateral. **b** Region-of-Interest

(ROI) analysis of mnemonic activity in the OFC and aVTC nodes. *: $p = 0.00081$ and $0.0031$ for OFC and aVTC, respectively, two-tailed unpaired $t$-test, Bonferroni-corrected. +: $p = 2.5 \times 10^{-7}$ and $4.5 \times 10^{-8}$ for OFC and aVTC, respectively, paired $t$-test against zero, Bonferroni-corrected. Error bars, sem. $n = 130$ and 100 for Ctrl and DREADD agonist conditions, respectively. Source data are provided as a Source Data file.

(Fig. 3a bottom, b left, and Supplementary Fig. 3a left, Supplementary Fig. 3b left, c for each monkey) as expected. A similar effect was observed for the signal in the mirror-symmetric site of the opposite hemisphere (Supplementary Fig. 3d left), consistent with the DREADD

expression in both hemispheres (see "Methods"). Note that OFC silencing also had a significant, memory-specific behavioral effect during the scanning sessions ($p < 0.007$, two-way ANOVA, interaction between factors of delay length and OFC silencing) (Supplementary

Fig. 3e). Within the OFC, the silencing effect locally extended to both the lateral and medial regions (Supplementary Fig. 3c), presumably reflecting the silencing at axon terminals[39] of locally projecting hM4Di-expressing neurons, as was consistently observed with both [¹¹C]DCZ-PET and histology (Supplementary Fig. 3f cyan and magenta).

In addition to those local activity changes around DREADD expression sites, OFC silencing also significantly attenuated the mnemonic signal in the aVTC ($p < 0.004$, vehicle vs. DREADD agonist) (Fig. 3a, b right and Supplementary Fig. 3a right, f right) in each of both monkeys (Supplementary Fig. 3b right, c), providing causal evidence of top-down aVTC regulation exerted by the OFC for object memory. Note that the aVTC node exhibited significant rs-fMRI connectivity with the OFC node (Supplementary Fig. 4b left, c). Furthermore, consistent with the observed rs-fMRI connectivity and previous anatomical studies[46,47], the anatomical connectivity from the DREADD-expressing OFC site to the aVTC was histologically confirmed (Supplementary Fig. 4d), suggesting that the observed top-down aVTC regulation by the OFC might be implemented at least partly via a direct anatomical pathway. In addition to the aVTC, significant remote effect of OFC silencing was also observed in the ACC ($p < 0.001$) (Supplementary Fig. 4a left), with which the OFC node exhibited significant rs-fMRI connectivity in addition to the aVTC (Supplementary Fig. 4b right, c). In contrast, significant mnemonic activity was observed in the IPS regardless of OFC silencing (Supplementary Fig. 4a middle), and appeared in the caudal entorhinal cortex (cEnt) only during OFC silencing (Supplementary Fig. 4a right), confirming that the DREADD agonist-induced reduction in the mnemonic activity was not caused by general whole-brain deactivation. In contrast to the aVTC and ACC nodes, neither IPS nor cEnt exhibited significant rs-fMRI connectivity with the OFC node (Supplementary Fig. 4c blue). Therefore, the spatial pattern of the remote silencing could be predicted from independent data of the rs-fMRI connectivity with the DREADD-expressing site. Collectively, the results causally demonstrated that focal silencing of the OFC node macroscopically disrupted mnemonic information flow to the aVTC along the ventral fronto-temporal network, which was accompanied by memory-specific behavioral impairment.

### OFC silencing microscopically attenuates stimulus-selective delay activity of individual aVTC neurons

To causally dissect more exact content and dynamics of mnemonic information flow along the ventral fronto-temporal network

microscopically at a cellular level, we next attempted to compare the activity of the same individual aVTC neurons between the conditions of before and during OFC silencing. To this end, we first identified the neuronal substrates of mnemonic fPET activation in the aVTC by conducting single-neuron recordings during the same task (3-s delay for both monkeys) at around the aVTC activation site in the same monkeys (Fig. 4a and Supplementary Fig. 5a). Some aVTC neurons exhibited stimulus-selective activity during both the cue and delay periods (Fig. 4b). These neurons were spatially clustered at a region spanning approximately 2-mm in diameter at the anterior region to the anterior tip of the anterior middle temporal sulcus (Fig. 4c and Supplementary Fig. 5b, c), consistent with the location of the fPET-identified mnemonic activation site (Fig. 3a top and Supplementary Fig. 3a right, c). Prominent stimulus selectivity during the delay period was characterized by significantly higher-than-baseline activity for the preferred stimulus ($p < 0.001$, paired $t$-test) (Fig. 4d) in each monkey (Supplementary Fig. 5d).

We next compared the activity of the same individual aVTC neurons between the conditions of before and during OFC silencing. Overall, 50 cue- and delay-selective neurons were recorded both before and after DCZ administration in 42 recording sessions. The numbers of trials recorded for the preferred stimulus before and after DCZ administration were $10.9 \pm 0.4$ and $11.3 \pm 0.5$ (mean ± sem), respectively. Consistent with the remote attenuation of mnemonic fPET signal in the aVTC node, OFC silencing with intravenous administration of DCZ during recordings significantly attenuated the delay activity of aVTC neurons specifically following the presentation of the preferred stimulus [$p < 0.001$, paired $t$-test, corrected for multiple comparisons, following two-way ANOVA, interaction between factors of OFC silencing and stimulus ($p < 0.001$)] (Fig. 5a, c, d magenta, and Supplementary Fig. 5f, g right for each neuron) in each monkey (Supplementary Fig. 5h, i). Of the total population of neurons that was analyzed (i.e., delay selective neurons recorded both before and during OFC silencing), 50% (25 of 50 neurons) exhibited a statistically significant reduction in the delay period activity for the preferred stimulus at the single-neuron level ($p < 0.05$, unpaired $t$-test). No single neuron showed the opposite effect (i.e., significant enhancement of the delay activity for the preferred stimulus). Note that these OFC silencing-induced changes in neuronal activity were accompanied by significantly deteriorated mnemonic behavioral performance during the recordings ($p < 0.002$, paired $t$-test) (Supplementary Fig. 5e).

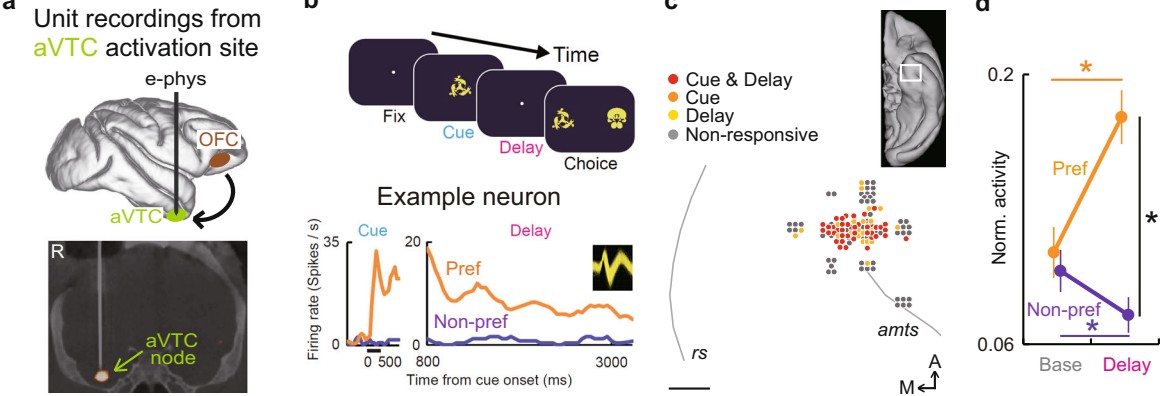

**Fig. 4 | fPET-guided electrophysiology in the aVTC node. a** Top, schema depicting fPET-guided electrophysiology from the aVTC node. Bottom, CT-image of electrode penetration in monkey 2, on which the fPET activity map was overlaid. R right. **b** Activity of an example neuron during the DMS task showing stimulus-selective responses during both the cue and delay periods. Black horizontal bars below the traces, cue period. Inset, spike waveforms of the neuron. **c** Spatial distribution of neurons showing stimulus-selective cue and delay activity in monkey 1 around the fPET activation site in the aVTC. Gray lines, fundus of sulci. rs rhinal

sulcus, amts anterior middle temporal sulcus. Scale bar, 1 mm. Inset, ventral view of the left hemisphere with white rectangle depicting the approximate map location. A anterior, M medial. **d** Delay activity of the aVTC neuronal population for preferred (orange) and non-preferred (purple) stimuli compared with the baseline. *: $p = 3.0 \times 10^{-9}$, 0.0012, and $1.2 \times 10^{-12}$ for Preferred, Non-preferred, and Preferred vs. Non-preferred, respectively, two-tailed paired $t$-test. Error bars, sem. $n = 50$ neurons showing significant stimulus selectivity during both cue and delay periods. Source data are provided as a Source Data file.

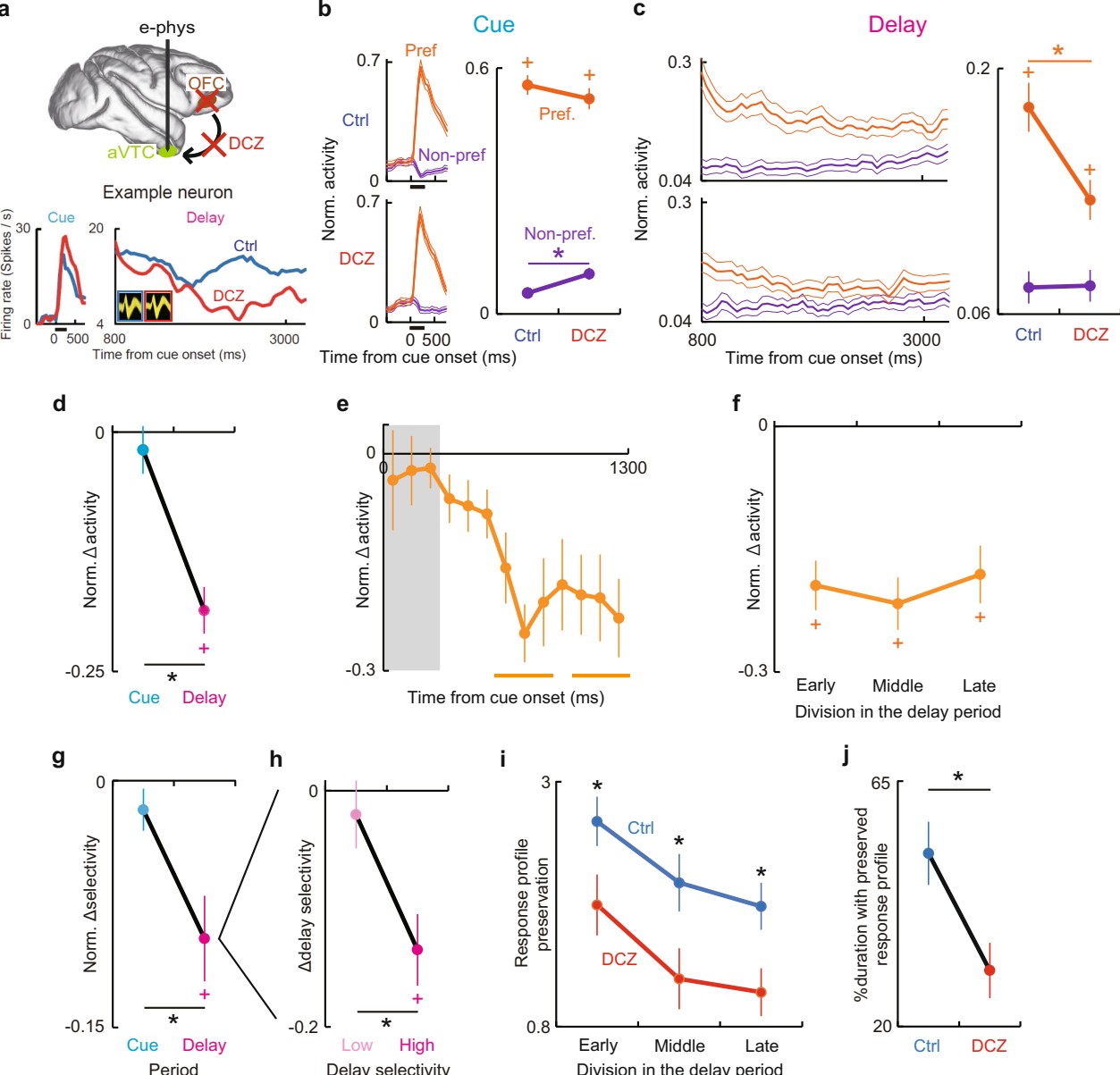

**Fig. 5 | Neuronal substrates of remote attenuation of aVTC mnemonic signal by OFC silencing. a** Activity of example aVTC neuron for preferred stimulus before (blue) and during (red) OFC silencing. Inset, spike waveforms before (blue) and during (red) OFC silencing. **b** Population activity of aVTC neurons during cue period for preferred (orange) and non-preferred (purple) stimuli before and during OFC silencing. *: $p = 0.00033$; paired $t$-test; +: $p = 1.2 \times 10^{-25}$ and $1.6 \times 10^{-18}$ for Ctrl and DCZ; Pref vs. Non-pref, paired $t$-test following two-way ANOVA with interaction between OFC silencing and Stimulus ($p = 0.0044$). Thick and thin traces, mean and mean ± sem. **c** Same as (**b**), but for delay period. *: $p = 9.0 \times 10^{-8}$; +: $p = 1.2 \times 10^{-12}$ and $3.2 \times 10^{-5}$ for Ctrl and DCZ; paired $t$-test following two-way ANOVA with interaction between OFC silencing and Stimulus ($p = 4.4 \times 10^{-8}$). Traces show neuronal activity before object presentation to touch, without eye movements for target search. **d** OFC silencing-induced activity changes for preferred stimulus. *: $p = 6.5 \times 10^{-6}$, paired $t$-test. +: $p = 3.4 \times 10^{-9}$, paired $t$-test against zero. **e** Latency of effect of OFC silencing on response to preferred stimulus. Gray rectangle, cue period. Orange

bars, $p < 0.05$; paired $t$-test, following one-way ANOVA ($p = 4.7 \times 10^{-7}$). **f** Sustained effect of OFC silencing on delay activity for preferred stimulus. *: $p < 1.0 \times 10^{-5}$, paired $t$-test. **g** OFC silencing-induced changes in stimulus selectivity. *: $p = 0.0035$, paired $t$-test. +: $p = 0.0011$, paired $t$-test against zero. **h** OFC silencing-induced changes in stimulus selectivity as a function of original selectivity during delay period. *: $p = 0.0085$, paired $t$-test. +: $p = 0.00033$, paired $t$-test against zero. **i** Preservation of cue representations during delay period before (blue) and during (red) OFC silencing. *: $p = 0.0026, 0.0058, 0.00022$ for Early, Middle, Late; paired $t$-test following two-way ANOVA with main effect of OFC silencing and Period ($p = 6.4 \times 10^{-6}$ and $1.5 \times 10^{-5}$). **j** Duration with significantly preserved cue representations before (blue) and during (red) OFC silencing. *: $p = 7.1 \times 10^{-5}$, paired $t$-test. Bonferroni-corrected for (**b–d**) and (**f–j**). $n = 50$ neurons showing significant stimulus selectivity during both cue and delay periods for (**b–g**) and (**i, j**), $n = 25$ for each of Low and High for (**h**). For (**b–j**), error bars are sem, statistical analyses are two-tailed, and source data are provided as a Source Data file.

In contrast, however, the preceding cue activity in the same neurons for the preferred stimulus did not significantly change following the same OFC silencing ($p > 0.5$) (Fig. 5a, b, d cyan) in each monkey (Supplementary Fig. 5h, i). The DCZ-induced delay activity reduction for the preferred stimulus did not simply reflect the time lapse or operation of administration, because this effect was significantly greater than that

induced by the vehicle administration ($p < 0.001$, unpaired $t$-test), which did not affect the activity (two-way ANOVA, main effect of OFC silencing, $p > 0.3$; interaction between factors of OFC silencing and stimulus, $p > 0.3$) (Supplementary Fig. 6a, b for each neuron). For the preferred stimulus, significant effect of OFC silencing appeared within 500 ms following cue offset (Fig. 5e) and was stably retained

throughout the delay period (Fig. 5f) in each of both monkeys (Supplementary Fig. 5j, k), but was not observed during the choice or baseline period ($p > 0.2$) (Supplementary Fig. 5l).

As a result, OFC silencing significantly disrupted the stimulus selectivity of aVTC neuronal activity specifically during the delay period, but not the cue period ($p < 0.002$ and $p > 0.1$ for the delay and cue periods, respectively; paired $t$-test, corrected for multiple comparisons; $p < 0.004$, delay vs. cue period) (Fig. 5g). Note that stimulus-selective delay activity would be required to retain the identity of the to-be-remembered object to correctly perform the task. Furthermore, this DCZ-induced selectivity reduction during the delay period was more prominent for neurons with higher stimulus selectivity during the delay period ($p < 0.001$ and $p > 0.8$ for neurons with high and low stimulus selectivity, respectively, paired $t$-test, corrected for multiple comparisons; $p < 0.009$, high vs. low selectivity) (Fig. 5h) (Supplementary Fig. 6c for each monkey). This was not the case for the cue period ($p > 0.1$ and $p > 0.9$ for high and low stimulus selectivity, respectively; $p > 0.3$, high vs. low selectivity; $p < 0.004$, delay vs. cue for high selectivity) (Supplementary Fig. 6d). These results suggest that OFC-regulated top-down modulation was specifically exerted on the aVTC neurons prominently encoding task-relevant mnemonic information.

We also investigated the effect of OFC silencing on the stimulus discriminability by applying the receiver operating characteristic (ROC) analysis. Before OFC silencing, the area under the ROC curve for discrimination between the preferred and non-preferred stimuli calculated for each neuron was not significantly different between the cue and delay periods as a population ($p > 0.1$, paired $t$-test) (Supplementary Fig. 6e). As a result of OFC silencing, the area under the ROC curve significantly more largely decreased during the delay period compared to the cue period ($p < 0.001$, two-way ANOVA, Interaction between factors of OFC silencing and period; $p < 0.001$, paired $t$-test) (Supplementary Fig. 6e, f). These results are consistent with the aforementioned results of delay-dominant impact of OFC silencing on the stimulus selectivity of aVTC neurons.

We next examined the effect of OFC silencing on the responses to each of all the tested stimuli. During the delay period, significant activity decrease was observed for the 1st- to the 3rd-preferred stimuli ($p < 0.001$, two-way ANOVA, Interaction between factors of OFC silencing and stimulus; $p < 0.003$, paired $t$-test, corrected for multiple comparisons) (Supplementary Fig. 6g, right). During the cue period, in contrast, only a significant response increase was observed for the non-preferred stimulus ($p < 0.001$, two-way ANOVA, Interaction; $p < 0.03$, paired $t$-test) (Supplementary Fig. 6g, left). When the OFC was silenced, relative representations of all the stimuli during the cue period were significantly less preserved throughout the delay period within each neuron ($p < 0.001$, two-way ANOVA, main effect of OFC silencing) (Fig. 5i), and the resultant duration with preserved relative representations became significantly shorter in each neuron ($p < 0.001$, paired $t$-test) (Fig. 5j). These results of OFC silencing-induced changes in the aVTC neuronal activity confirm the remote aVTC silencing macroscopically observed in the whole-brain fPET (Fig. 3), and further provide more detailed characterization of how the top-down input from the OFC dynamically and specifically supports the retention of mnemonic representations in individual aVTC neurons.

### Activity changes of the aVTC neurons by OFC silencing mirrors that in mnemonic error trials before silencing

If the observed top-down aVTC regulation is relevant to mnemonic performance, the strength of delay activity for the preferred stimulus in a given trial should predict the resultant mnemonic performance of the trial in intact condition. Trial-based analysis of aVTC neuronal activity for the preferred stimulus before OFC silencing indeed revealed significantly decreased activity in mnemonic error trials with wrong choices compared to those in correct trials; furthermore, this

was specifically true during the delay, but not cue, period ($p < 0.02$ and $p > 0.3$ for the delay and cue period, respectively; paired $t$-test, corrected for multiple comparisons; $p < 0.002$ for cue vs. delay) (Fig. 6a, b and Supplementary Fig. 7a for each neuron). The degree of delay activity decrease in mnemonic error trials could not be predicted from that of the preceding visual response (cue activity) for each neuron ($p > 0.5$) (Supplementary Fig. 7b), and was not observed following presentation of the non-preferred stimulus ($p > 0.7$). The activity reduction preceding mnemonic errors emerged significantly within 500 ms following cue offset (Fig. 6d), and retained throughout the delay period (Fig. 6c). Therefore, these features of activity change in mnemonic error trials observed for aVTC neurons mirrored the aforementioned impact of OFC silencing. This suggests that the activity of aVTC neurons during the delay, but not the cue, period for the preferred stimulus in a given trial predicts the mnemonic performance of the trial, and that OFC silencing switches the activity mode of aVTC neurons to that in mnemonic error trials, leading to the observed memory impairment.

If this is the case, i.e., if mnemonic error-predicting decrease in the delay activity of aVTC neurons is derived from the decrease in the top-down signal from the OFC, then OFC silencing should minimize the mnemonic performance-related difference in aVTC delay activity for the preferred stimulus. The obtained results confirmed this specific prediction ($p > 0.8$ and $p > 0.5$ against zero during OFC silencing for delay and cue periods, respectively; $p > 0.1$ for cue vs. delay; $p < 0.02$ for control vs. OFC silencing during the delay period, unpaired $t$-test) (Fig. 6e, f and Supplementary Fig. 7c for each neuron). Note that if the performance dropped to around 75% overall during OFC silencing, then many of the correct trials might simply be the result of guessing; it would then be unclear whether the neuronal activity was expected to be so different between the correct and error trials—or there might be no room for a difference to be observed. However, the electrophysiological experiments were conducted with a 3-s delay condition. In this condition, the correct rate was above 90% on average even after OFC silencing, and the average difference in performance between the conditions with and without silencing was less than 5% (although this difference was statistically significant, as shown in Supplementary Fig. 5e). Taken together, therefore, our results support the idea that dynamic top-down regulation by the OFC along the ventral fronto-temporal network implements short-term object memory via specific retention of the performance-predictive delay activity of individual aVTC neurons encoding the to-be-remembered object (Fig. 6g).

## Discussion

Using a multifaceted but convergent approach of combining chemogenetic fPET/electrophysiology during a task with fMRI connectivity during rest, we causally demonstrated at both macro- and micro-scales that the well-established mnemonic object representations in the primate aVTC are implemented via dynamic top-down regulation by the OFC along the ventral fronto-temporal network and is predictive of trial-by-trial mnemonic performance. The present study provides a comprehensive answer to the following long-lasting questions: in the context of object memory, where in the brain top-down regulates the aVTC? Which component of the aVTC neuronal activity is top-down modulated at which timing? And is the modulation relevant to mnemonic performance?

Current results extend the conventional view on the aVTC as the apex of visual object processing[20–22] to that it dynamically interacts with the OFC especially when visual information about familiar objects has to be processed for functions beyond perception. In contrast to the bottom-up perceptual representations of external stimuli, mnemonic representations in the aVTC were found to require the top-down regulation by the OFC, both of which are consistent with observed differential behavioral impacts of OFC silencing depending on the delay length. Furthermore, changes in the aVTC neuronal

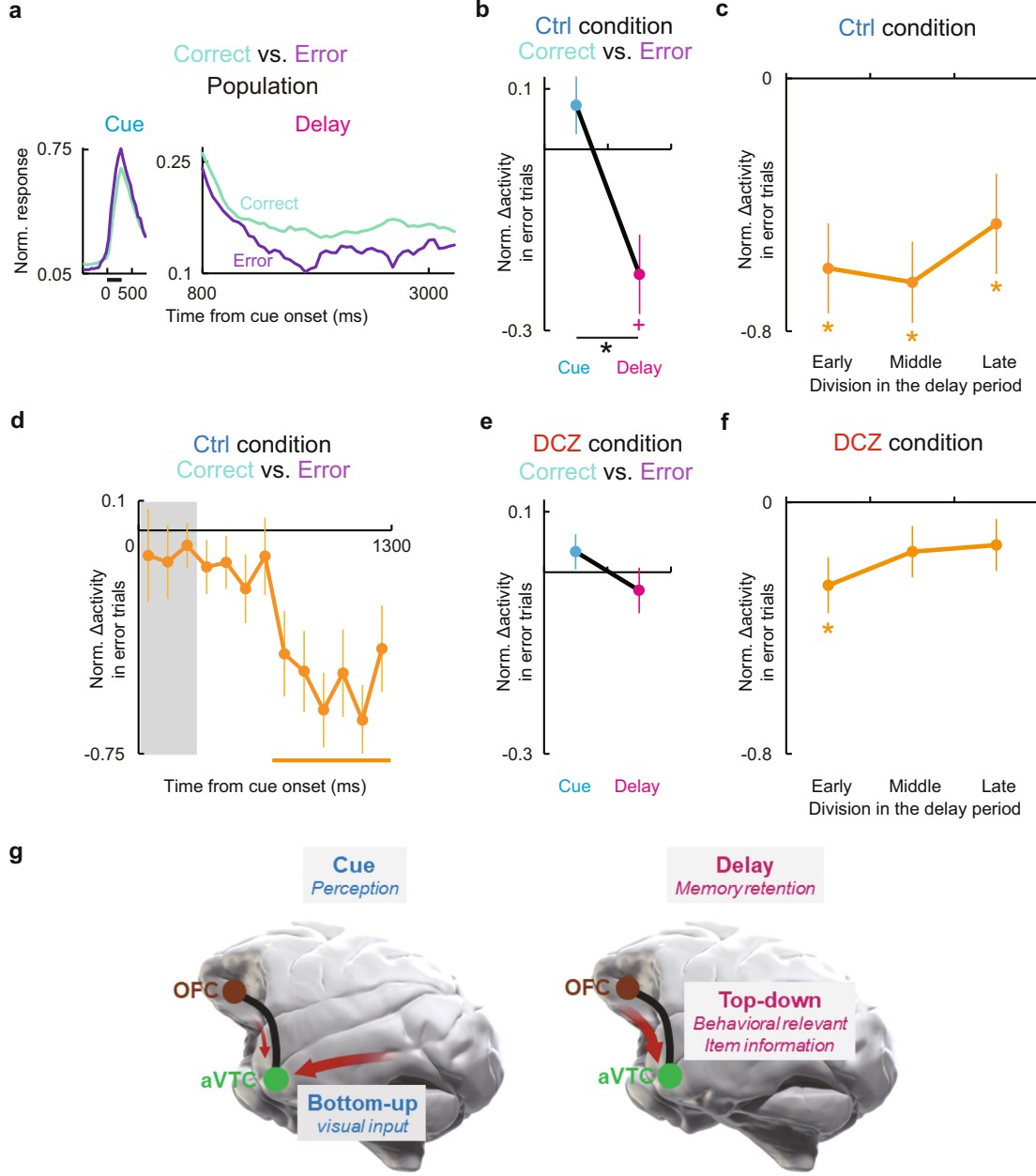

**Fig. 6 | Mnemonic performance predictability of aVTC neuronal delay activity and schema of the ventral fronto-temporal network operation for visual object memory. a** Population activity of aVTC neurons in correct (light green) and error (light purple) trials during the cue (left) and delay (right) periods. **b** Mnemonic error-predicting normalized activity change during cue (cyan) and delay (magenta) periods before OFC silencing. *: $p = 0.0017$, paired $t$-test. +: $p = 0.012$, against zero, paired $t$-test, Bonferroni-corrected. **c** Persistence of activity change in mnemonic error trials before OFC silencing. *: $p = 0.0016$, $0.00027$, and $0.045$ for Early, Middle, and Late, respectively, paired $t$-test against zero, Bonferroni-corrected. **d** Latency of activity change in mnemonic error trials against cue onset. Gray rectangle, cue period. Orange bar, $p < 0.05$, paired $t$-test against zero, following one-way ANOVA ($p = 4.9 \times 10^{-6}$). **e** Same as (**b**), but during OFC silencing. **f** Same as (**d**),

but during OFC silencing. *: $p = 0.015$, paired $t$-test against zero, Bonferroni-corrected. $n = 23$ and $41$ mnemonic error trials (21 and 28 neurons showing significant stimulus selectivity during both cue and delay periods) in the control (**a**–**d**) and DCZ administration (**e**, **f**) conditions, respectively. For (**b**–**f**), error bars are sem, statistical analyses are two-tailed, and source data are provided as a Source Data file. **g** Schema of the ventral fronto-temporal network operation for visual object memory. During the cue period (left), aVTC neurons are primarily activated by bottom-up visual input for perception, during which the effect of top-down regulation remains minimal. During the delay period without visual input (right), top-down signal from the OFC prominently upregulates the aVTC neurons carrying behaviorally relevant object information to maintain the representation of the presented cue.

activity during OFC silencing was found to mirror that in mnemonic error trials before silencing, suggesting behavioral relevance of the observed top-down regulation, consistent with previous macaque studies showing that the aVTC inactivation induced object memory deficits[1–3,5].

Meanwhile, the primary prefrontal node for object memory via interaction with the VTC has been widely believed to be the LPFC in

primates[19,50,51]; however, unbiased whole-brain functional imaging has not been conducted in monkeys to identify the fronto-temporal network for object memory. Most of the human imaging data have inevitably incorporated the effect of verbalization, and thus the core circuit purely for object memory has been rarely extracted (but see ref. 52). The current study identified the OFC as a critical partner of the aVTC for object memory via unbiased and multimodal

causal neuroimaging. Previous studies have shown that among the prefrontal cortex (PFC), the OFC is one of subregions with which the aVTC anatomically most densely connected[46,47,53], consistent with the current findings. We also demonstrated specifically where, when, and how the mnemonic signal derived from the OFC is leveraged in another brain region at both spatial scales. Although the OFC has rarely been considered an indispensable node for object memory, its causality in macaques[4] and activation in humans[52,54–57] have been shown for object memory. However, the brain-wide network in which the OFC is embedded for the mnemonic function has not been examined. The present results support a recent view on the role for the OFC in tracking behaviorally relevant latent task states[58], and extend this account to a large-scale network view that the OFC implements object memory via dynamic top-down regulation of mnemonic object representations in the aVTC.

Although individual neurons within the fPET-identified aVTC node exhibited stimulus-selective delay activity with a higher firing rate than the baseline and the trial-by-trial predictability of mnemonic performance, these traits are uncommon in the whole of the VTC[19]. It has been shown that the aVTC is more critical than the posterior VTC for object memory[2,3] and is anatomically more densely connected with the OFC[46,47], consistent with the current results. Whole-brain functional imaging would thus be an indispensable tool for unbiased and precise mapping of critical network nodes to meaningfully guide subsequent closer examinations, including focal perturbation[24,25,27,28,59–61] and neuronal recordings[29–31].

In a DMS task composed of several epochs, OFC silencing predominantly affected the delay activity of aVTC neurons. Although massive anatomical connectivity exists bidirectionally between these regions[46,47] in both macaques and humans[62,63], the current results suggest that the functional connectivity between these regions would not be static, but rather dynamic: information flow between these regions would be weak or primarily bottom-up during the cue period with massive visual input to drive the aVTC, whereas the top-down influence appeared predominantly during the delay period without the visual input (Fig. 6g). Such dynamic functional connectivity between neurons within[64–66] or across[67–70] brain areas depending on the task phase or processing has been shown in previous studies. Note that the causally observed top-down information flow from the OFC to the aVTC might be part of a more symmetric recurrent interaction between them. Regarding the pathway which implements the above functional interactions, although anatomical connectivity from the DREADD-expressing OFC site to the aVTC was identified in the present study, we did not directly test whether the observed causal influence exerted by the OFC on the aVTC was implemented by the identified anatomical pathway. Therefore, causal demonstration of the role for anatomical OFC to aVTC pathway using chemogenetic synaptic sliencing[39] or other techniques would be an important issue for future work.

In the current study, visual stimuli were repeatedly presented throughout the training and data acquisition, and were thus highly familiar for the animals even at the beginning of the data acquisition. This procedure would have made a specific population of aVTC neurons more selectively and strongly responsive to the set of stimuli used in the task[71–73], consistent with the observed neuronal responses in the aVTC activation site. Likewise, OFC activity in object recognition memory task has been shown to be more prominent for familiar stimuli than for novel ones[54,55] (see also refs. [52,74]). Familiarity is known to improve short-term object memory in humans owing to the neural system for long-term semantic memory[75], which lies in the aVTC[76]. Meanwhile, familiarity and repeated presentations of to-be-remembered objects would increase the interference among them, and it might recruit the OFC more prominently[55]. Behavioral vulnerability of OFC damage to such mnemonic interference has been demonstrated in humans[77], and it might be explained by its role in

tracking latent task state[58]. Therefore, the currently observed operation of the ventral fronto-temporal network including the top-down regulation of aVTC by the OFC and their causality on the mnemonic performance likely depended on the familiarity of the visual stimuli used in the task. However, the present study did not test novel objects, and the network difference in the perceptual/mnemonic representations between the novel and familiar objects remains elusive. Similarly, although the current study focused on the short-term memory, the ventral fronto-temporal network likely plays important roles in various cognitive functions related to visual object processing[78]. For example, because the aVTC has been well established as a storehouse of visual long-term memory[20,21,79], similar network might be used in the retrieval of visual long-term memory, especially when the context-dependent, controlled flexible retrieval is required[17,80], which would be an intriguing possibility to be examined in future work.

In the present study, we leveraged PRIME-DE—a publicly available database of macaque rs-fMRI[45]—to analyze the functional connectivity among the fPET activation sites. The database specifically we used included rs-fMRI data acquired from a population of twenty macaques, with over 17 h-long recording data in total[44]. The calculated functional connectivity was thus more reliable than that obtained from two to four macaques in standard research settings, offering 'quasi-normative' connectome. Such invaluable connectome data can be effectively combined with a variety of data modalities other than fMRI. In the present study, it provided the resting-state functional network in which fPET activation sites were embedded, which cannot be obtained from the fPET data alone. Based on the ventral fronto-temporal network identified as a circuit for short-term object memory by combining fPET and rs-fMRI connectivity, we further conducted chemogenetic silencing and evaluated its remote effects with both fPET and electrophysiology. The connectome data could also predict whether mnemonic activity in a given remote region was attenuated by OFC silencing or not, complementarily supporting the results of the chemogenetic fPET. This combination of data obtained with multiple approaches, but in a manner closely related and complementary to each other, allowed us to comprehensively and causally understand the operation of the ventral fronto-temporal network for object memory in primates, which has not been examined previously.

Note that the identified ventral fronto-temporal network would implement object memory in concert with other regions including the LPFC, posterior parietal cortex, and subcortical regions as previously reported[81–83] and partly identified in the current fPET (Figs. S1c and S4a). This might at least partly explain why the behavioral impact of OFC silencing was partial within the range of delay length currently tested, in spite of relatively strong physiological local effect observed in fPET. Likewise, mnemonic performance-predictive delay activity change of aVTC neurons was minimized during OFC silencing, implying that the trial-by-trial mnemonic performance in this condition might have depended on activity fluctuations of other contributing networks. Note that, however, the current OFC silencing was not the whole, but was restricted to the fPET activation sites, which might have also limited the behavioral impact. Nevertheless, highly restricted current OFC silencing still exerted the delay length-dependent behavioral impact; this suggests that other networks were not enough to perform this task in longer delay conditions, highlighting the importance of the identified ventral fronto-temporal network for object memory especially when longer retention was required. In contrast to the OFC, our fPET during a DMS task did not show prominent LPFC activation, consistent with previous lesion studies in macaques showing that the OFC lesion induced more severe deficit in object recognition memory compared with lesions to the dorsolateral or ventrolateral PFC[4,7,84]. However, the current negative results in neuroimaging might have derived from inevitable factors including the granularity of activation[85], and previous studies have suggested that the LPFC also plays a role in short-term object memory

through interaction with the posterior parietal[86] and/or temporal cortices[16,19,50]. In particular, the currently observed mnemonic activation site in the IPS has been shown to be functionally connected with the dorsolateral PFC[87], constituting a fronto-parietal network for both spatial and non-spatial visual short-term memory[86,88]. Current results of both chemogenetic fPET and rs-fMRI connectivity consistently suggest that this dorsal network is functionally relatively independent from the ventral fronto-temporal network as has been implied based on the anatomical connectivity[53]. Given the critical role for the LPFC in selection and monitoring of both external stimuli and internal representations[89], more prominent LPFC activation would be expected for task paradigms with intervening distractors or retention of multiple objects[5,90]. Another point to be considered is that the current fPET was conducted in a block design, in which trials with long- and short-delay were temporally separated from each other (but the current block was not explicitly informed to monkeys), which might have affected the pattern of activity.

Because OFC has traditionally been characterized as a critical node for object evaluation or value-based decision making[91–93], one might argue that the current OFC activation might have related to object values and the behavioral effect of OFC silencing might have resulted from a loss of value-related information. However, both the quality and quantity of the reward in each trial were the same irrespective of different delay lengths, for which both differential OFC activity in the fPET and differential behavioral impact of OFC silencing were observed; therefore, neither the observed OFC activation nor the behavioral impact of its disruption was likely to be related to object values. Likewise, it seems also unlikely that impaired mnemonic performance was attributable to a reduction in general motivation due to OFC silencing, because the rate of inappropriate lever release before the onset of the choice period was unchanged and did not correlate with the mnemonic performance during the OFC silencing. Furthermore, significant activation of other value- or reward-related brain regions, such as the amygdala or ventral striatum, were not observed in our fPET. However, given that OFC neurons show a delay activity related to the subjective value of the forthcoming reward in a delayed response task[94], it remains an intriguing question what exactly OFC neurons encode in mnemonic tasks.

The observed effects of OFC silencing were unlikely to have been caused by unwanted off-target actions of DREADD agonists for the following four reasons. First, the administered doses of DREADD agonists were within the range used in previous macaque studies[27,35,38], which have been shown to avoid unwanted off-target actions in terms of both whole-brain activity and behavior[38,95]. Second, behavioral testing indicated that the effects of DREADD agonists were not general but were specific to the conditions with mnemonic demand; the agonists had no effect on the performance of a control task that required the same sensorimotor function, decision making, and rule-based behavior but did not require mnemonic function. Critically, the mnemonic behavioral impact of an agonist was not observed when tested before DREADD expression, even in the most demanding condition (i.e., 9-s delay). Third, chemogenetic fPET indicated that although the ventral fronto-temporal network and ACC were downregulated by OFC silencing, the IPS and cEnt were not; furthermore, this pattern of remote silencing was not random, but rather predictable from the rs-fMRI connectivity with the silenced OFC node. And fourth, chemogenetic electrophysiology revealed that the effect of OFC silencing on the activity of individual aVTC neurons was predominantly observed for the mnemonic signal during the delay period following presentation of the preferred stimulus; perceptual signal during the cue/choice periods for the same stimulus, the delay activity for the non-preferred stimulus, or the pre-cue baseline activity of the same neurons remained unchanged.

Critically, chemogenetic silencing allowed less-invasive and repetitive silencing of a specific neuronal population, providing an invaluable opportunity to directly link macro- and micro-scale causal evidence for top-down mnemonic regulation within the same monkeys for the first time. Our unique approach enables unbiased, comprehensive, and causal discoveries of critical network operations underlying given brain functions, irrespective of prevailing and influential models or hypotheses. Specifically, because the currently focused short-term object memory serves as a fundamental component of many other cognitive abilities, the identified network operations would be involved in various cognitive functions. Furthermore, the current approach would be widely applicable to causally elucidate the multiscale network operations underpinning cognition and emotion in primate models, which will aid in the translational understanding of corresponding functions and their disruption in the human brain.

## Methods
### Subjects
Two macaque monkeys (*Macaca mulatta*, male, weight: 6–8 kg, age: 5–10 years) were used for the experiments. (see *rs-fMRI connectivity analysis* for an additional population of 20 macaques in PRIME-DE open database[45]). Inhibitory DREADD (hM4Di) was expressed in the bilateral OFC of both monkeys via injections of AAV vectors encoding hM4Di. Both monkeys participated in localizer task fPET, fPET-guided hM4Di transduction, DCZ-PET for in vivo visualization of the expressed hM4Di, chemogenetic OFC silencing concurrent with behavioral tests/ fPET scans/electrophysiology. All experimental protocols were conducted in full compliance with the Guide for the Care and Use of Laboratory Animals (National Research Council of the US National Academy of Sciences) and were approved by the Animal Ethics Committee of the National Institutes for Quantum Science and Technology.

### Viral vector production
Viral vectors (AAV2-CMV-hM4Di and AAV2-CMV-AcGFP) were produced as in our previous studies[27,35] using a helper-free triple transfection procedure and purified by affinity chromatography (GE Healthcare). Viral titers were determined by quantitative polymerase chain reaction using Taq-Man technology (Life Technologies). The transfer plasmid was constructed by inserting a complementary DNA fragment and the WPRE sequence into an AAV backbone plasmid (pAAV-CMV, Stratagene).

### AAV vector injection into the mnemonic fPET activation site in the OFC
For both monkeys, a mixture of AAV2-CMV-hM4Di and AAV2-CMV-AcGFP ($2.2 \times 10^{13}$ and $2.3 \times 10^{13}$ genome copy/ml for monkey 1; $1.3 \times 10^{13}$ and $2.3 \times 10^{13}$ genome copy/ml for monkey 2) for the neuron-specific expression of hM4Di and GFP was injected into the OFC activation site identified with fPET during the DMS task. The combination of AAV2 and CMV promoter has been reported to enable neuron-specific expression of target proteins in macaques[49]. The AAV vectors used in these monkeys were the same as those used in our previous chemogenetic studies of behavior[35] and fMRI/behavior[27]. To determine the stereotaxic coordinates of AAV vector injection sites and the depth from the skull surface, computed tomography (CT) images (acquired using Accuitomo170, J. MORITA CO.) of the skull for each monkey were coregistered to the structural MR images, upon which the co-registered fPET activation map (see below) was overlaid, using PMOD image analysis software (PMOD Technologies Ltd.). On the day of AAV vector injection surgery, the monkeys were initially anesthetized with ketamine (5–10 mg/kg) and xylazine (0.2–0.5 mg/kg) complemented with atropine, and the anesthesia was then maintained with approximately 1 to 2% isoflurane. Vital measures (heart rate, SpO$_2$, rectal temperature, and end-tidal CO$_2$) were continuously monitored throughout the surgery. Injections were conducted using a 10-µl syringe with a 30-gauge needle (Model 1701RN, Hamilton). The syringe was mounted onto a motorized microinjector (Legato130, KD

Scientific or UMP3T-2, WPI) that was held by a manipulator (model 1460, David Kopf) on the stereotaxic frame. After retracting the skin and galea, a burr hole (8-mm diameter) and the dura mater (approximately 5 x 5 mm) were opened for each hemisphere. The injection needle was then inserted into the brain, and moved down to 0.5 mm below the target depth (i.e., the middle point of the cortex at the OFC activation site), and then kept stationary for 5 min. The needle was then pulled up 0.5 mm, and the injection was started at 0.1 µl/min. Following each injection (2 and 3 µl for monkeys 1 and 2, respectively), the needle was slowly pulled up following a 15-min wait to prevent backflow. The viral vector injections were conducted in both hemispheres. The injection site in the opposite hemisphere to the OFC activation site with the largest z-value was determined as its mirror-symmetric site.

### Histology and immunostaining
For histological analyses, monkeys were immobilized with ketamine (10 mg/kg, intramuscular), and then deeply anesthetized with an overdose of sodium pentobarbital (80 mg/kg, intravenous) and transcardially perfused with saline at 4 °C, followed by 4% paraformaldehyde in 0.1 M phosphate-buffered saline (PBS), pH 7.4. The brains were removed from the skull, postfixed in the same fresh fixative overnight, saturated with 30% sucrose in phosphate buffer at 4 °C, and then cut serially into 50 µm-thick sections on a freezing microtome. For the visualization of immunoreactive signals of GFP, a series of every 6th section was immersed in 1% skim milk for 1 h at room temperature before being incubated overnight at 4 °C with rabbit anti-GFP monoclonal antibody (1:1000; Thermo Fisher Scientific) in phosphate-buffered saline containing 0.1% Triton X-100 and 1% normal goat serum for 2 days at 4 °C. The sections were then incubated in the same fresh medium containing biotinylated goat anti-rabbit IgG antibody (1:1000; Jackson ImmunoResearch) for 2 h at room temperature, followed by avidin-biotin-peroxidase complex (ABC Elite, Vector Laboratories, Burlingame) for 1.5 h at room temperature. For visualization of the antigen, the sections were reacted in 0.05 M Tris-HCl buffer (pH 7.6) containing 0.04% diaminobenzidine (DAB), 0.04% $NiCl_2$, and 0.003% $H_2O_2$. The sections were mounted on gelatin-coated glass slides, air-dried, and cover-slipped. A part of other sections was Nissl-stained with 1% cresyl violet. Images of the sections were digitally captured using an optical microscope equipped with a high-grade charge-coupled device (CCD) camera (Biorevo, Keyence) and a viewer software (NDP.view2, Hamamatsu Photonics K.K.).

### In vivo PET visualization of DREADD expression
Approximately 7 weeks following the virus injection surgery, and before the functional investigations of hM4Di, the expression level and spatial distribution of the transduced hM4Di were first examined in vivo as in previous studies[27,38,39] using PET with a radio-labeled DREADD agonist ([11C]DCZ) as a tracer, which specifically binds to the expressed DREADD[38]. PET scans were conducted using a microPET Focus 220 scanner (Siemens Medical Solutions USA, Malvern). Anesthesia and vital monitoring were conducted as described above for the virus vector injection surgery. Transmission scans were performed for approximately 20 min with a Ge-68 source. Emission scans were acquired in three-dimensional list mode with an energy window of 350–750 keV after intravenous bolus injection of [11C]DCZ (344.8–369.6 MBq). Emission data acquisition lasted 90 min. PET images were reconstructed with filtered back-projection using a Hanning filter cut-off at a Nyquist frequency (0.5 mm⁻¹). To estimate the expression level of transduced DREADD in vivo, the specific binding of [11C]DCZ was calculated as regional binding potential relative to non-displaceable radioligand ($BP_{ND}$) using PMOD with an original multilinear reference tissue model (MRTMo)[38]. The resultant $BP_{ND}$ map was coregistered to the structural MRI of the same individual using PMOD. The structural MRI was obtained under anesthesia [immobilized by ketamine (5–10 mg/kg)

and xylazine (0.2–0.5 mg/kg) complemented with atropine, and maintained with continuously intravenous injection of propofol (0.2–0.6 mg/kg/min)] using a 7-T MRI scanner (Magnet: KOBELCO/JASTEC; Spectrometer and console: AVANCE-I, Bruker Biospin) with a volume resonator (Takashima) and a four-channel phased array receiver coil (Rapid Biomedical) via fast low angle shot (FLASH) sequence [scan parameters: repetition time (TR), 660 ms; echo time (TE), 13.37 ms; flip angle, 20°; field of view (FOV), 150 × 150 mm; matrix size, 512 × 512; slice thickness, 1 mm; no spatial gap between slices; four sets of 24 coronal slices covering the whole brain].

### Behavioral task
Two monkeys were trained with a DMS task (Fig. 1b), which was controlled by Inquisit (Millisecond Software). When the background color of the touch screen (Elo TouchSystems, Tyco Electronics) placed in front of the monkey chair was dark blue, the task started if the monkeys touched the lever in front of the body. When the monkey touched the lever, a 500-ms pre-cue baseline period started with the appearance of a fixation point (approximately 0.5° × 0.5° in size) at the center of the dark blue background. Following the baseline period, a cue object (randomly chosen from a pool of 11 yellow-colored Fourier descriptors[65,96]; approximately 5.5° × 5.5° in size) appeared at the center of the screen (under the fixation point) for 300 ms. This was followed by the delay period, during which the cue object disappeared. At the choice period following the delay period, two different objects—one of which was the same as the cue object and the other of which was a distractor—appeared on the right and left sides of the screen (approximately 10° from the screen center), which required the monkey to release the lever and choose one of the objects by touching the screen. If the monkey correctly chose the object identical to the cue stimulus, a drop of juice was delivered, and only the chosen stimulus was further presented for 500 ms before the inter-trial interval (ITI) with a blank screen (black). Otherwise, if the monkey touched the distractor or released the lever before the choice period, then the ITI started just after the screen touch or the lever release without the reward delivery. The background color of the monitor then changed to dark blue following the ITI, signaling that the monkey can start the next trial by touching the lever. A lever release before the choice period was treated as task refusal, and the refusal rate was calculated as a proxy of each animal's general motivation. Sessions of behavioral testing with DREADD agonist or vehicle administration after AAV injections were conducted following in vivo confirmation of DREADD expression with [11C]DCZ-PET. DREADD agonist [0.1 mg/kg DCZ or 3 mg/kg CNO; the same concentrations with previous studies for both DCZ[27,97] and CNO[35,98]] or the corresponding vehicle was administrated intravenously 15 min before starting the first trial in each behavioral session. Sessions with administration of a DREADD agonist (8 sessions with DCZ and 10 sessions with CNO for monkeys 1 and 2, respectively) or vehicle (8 and 10 sessions for monkeys 1 and 2, respectively) were conducted separately. For monkey 2, additional 5 and 12 sessions were conducted with administrations of DCZ and vehicle, respectively. In addition, 5 and 7 sessions were conducted before DREADD transduction for monkey 1 with administration of CNO and vehicle, respectively.

### fPET scanning during the DMS task
To localize brain regions that exhibit the activity related to short-term visual object memory, [15O]H₂O PET (fPET), which measures regional cerebral blood flow during task execution, was conducted after the animals were trained for the DMS task. Prior to scanning sessions, non-magnetic head post was surgically implanted in each monkey under anesthesia and vital monitoring described for the virus vector injection surgery. In a scanning session, scans for short- (0.3 s) and long- (4 and 3 s for monkeys 1 and 2, respectively) delay condition were alternated without explicit cue informing the current task condition to monkeys. Monkeys performed the task in a vertical primate chair located under

the quasi-vertical PET scanner (SHR-7700, Hamamatsu Photonics K.K.)[41,42] (Fig. 1a, leftmost). The emission scan was conducted in the two-dimensional mode with an in-plane resolution of 2.6 mm in full width at half maximum (FWHM). For each emission scan, monkeys began to perform the task 2 min before the intravenous injection of [$^{15}$O]H$_2$O. Scanning started when a radioactivity count in the brain was detected (>30 kcps) following bolus intravenous injection of $^{15}$O-labeled water (approximately 1.1 GBq) via the crural vein using an automated water generator (A&RMC, Melbourne, Australia) (approximately 35 s after the injection start), and ended 2 min after the scan start (12 × 10-s frames). During the scan, each monkey performed approximately 20 trials of the DMS task in short- or long-delay conditions. Task durations in the two conditions with different delay lengths were adjusted to make them to be the same using the duration of the ITI. At the end of each scan, the task was temporally terminated. The next scan started when the radioactivity counts in the field of view dropped below 5% of the minimum requirement for scanning (<10 kcps). The resultant inter-scan interval was approximately 10 min. Scans after hM4Di transduction were conducted following in vivo confirmation of DREADD expression with [$^{11}$C]DCZ-PET. DREADD agonist (0.1 mg/kg DCZ and 3 mg/kg CNO for monkeys 1 and 2, respectively) or the corresponding vehicle were administered intravenously 15 min before starting the first scan in each scanning session. In total, 13 (6 and 7 for monkeys 1 and 2, respectively) sessions were conducted before hM4Di transduction; 24 (13 and 11 for monkeys 1 and 2) and 19 (8 and 11 for monkeys 1 and 2) sessions were conducted following hM4Di transduction for vehicle and DREADD agonist conditions, respectively. At the end of each scanning session, a 50-min transmission scan was obtained with a rotating $^{68}$Ge-$^{68}$Ga pin source to evaluate the relative attenuation factor for image reconstruction[41,42].

## Neuronal recordings at the fPET activation site in the aVTC

Following all of the aforementioned fPET scanning sessions after hM4Di transduction, a recording chamber was surgically implanted in each monkey under anesthesia and vital monitoring described for the virus vector injection surgery to cover the area enclosing the fPET activation site in the aVTC (left and right hemispheres in monkeys 1 and 2, respectively). Single-unit recordings were conducted during the DMS task with a delay period of 3 s. A tungsten electrode, manufactured in-house, was vertically lowered to the aVTC with an oil-driven manipulator (MO-95C, Narishige) following punctuation of the dura matter with a stainless-steel guide tube. The coordinate and depth of the electrode were examined with CT scan, and was compared with the fPET activation site in the aVTC (Fig. 4a and Supplementary Fig. 5a; see "Analysis of fPET data"). Neuronal activity was recorded using a TDT recording system (PZ5-64 NeuroDigitizer and RZ2 processor, Tucker Davis Technologies) with high- and low-pass filters of 500 Hz and 5 kHz, respectively at a sampling rate of 20 kHz, and the recorded data were digitally stored in a PC with a set of task and behavioral parameters. Neuronal recordings to examine the effect of OFC silencing were conducted following the mapping of neurons showing mnemonic delay activity in the aVTC. A "hot spot" of mnemonic neurons was identified in a region of approximately 2-mm diameter spanning along the ventral surface of the anterior temporal cortex, in the vicinity of fPET activation site (Fig. 4c and Supplementary Fig. 5c). In each recording session, following data acquisition in the intact condition, DCZ (0.1 mg/kg, the same concentration as that used in behavioral tests and fPET scans) was intravenously administered. After waiting for approximately 5 min, data were acquired from the same neurons in the OFC silencing condition. Neuronal activity was recorded for approximately 10 or more trials for each stimulus both before and after DCZ administration. To control for the effects of the operation of DCZ administration and time lapse, vehicle administration was conducted in some sessions, and recordings from the same neurons were conducted both before and after the administration. In some of these

sessions, DCZ was further injected after data acquisition in the vehicle condition, in which the data acquired following vehicle administration was treated as the control condition for the DCZ administration.

## Analysis of fPET data

For each fPET scan, emission data were summed for the first 60 s following scan start and were reconstructed using filtered back projection with a 4-mm Hanning filter to obtain fPET images representing a relative spatial distribution of rCBF[41]. The reconstructed fPET images were realigned via statistical parametrical mapping (SPM8) software on MATLAB (MathWorks) before being normalized using the following protocol. Individual structural MRIs were linearly and non-linearly registered to the Yerkes19 macaque template[99] using FMRIB's linear registration tool (FLIRT) and FMRIB's nonlinear registration tool (FNIRT) implemented in FSL software (FMRIB's Software Library, http://www.fmrib.ox.ac.uk/fsl)[100]. Realigned fPET images were normalized to the template using structural MRI-to-template matrices. Using SPM8, normalized images were then spatially smoothed (FWHM, 3 × 3 × 3 mm). The images were masked with the mask obtained from the template. $Z$-values of memory-related activity for both monkeys for each voxel were then calculated from the activity difference between the long and short delay conditions using the analysis of covariance considering the variance of global radioactivity for each scan. For across-subject analysis, fixed effect model was adopted as in previous fMRI studies in macaques[101–103], which was supplemented with single-subject analysis[101]. The statistical threshold for voxels with significant mnemonic activation was set at $p = 0.005$, uncorrected for multiple comparisons ($z = 2.58$), which was akin to that in previous fPET studies in macaques[42,104]. The obtained statistical maps were overlaid on the template image to visualize the activation using the Connectome Workbench[105]. To visualize global activation patterns, statistical maps were rendered on the template cortical surface[106] using the Connectome Workbench. Before hM4Di transduction, 64 (34 and 30 for monkeys 1 and 2) and 68 (39 and 29 for monkeys 1 and 2) scans were included in the analysis as the data in the long- and short-delay conditions, respectively. For vehicle administration following hM4Di transduction, 130 (71 and 59 for monkeys 1 and 2) and 128 (69 and 59 for monkeys 1 and 2) scans were included in the analysis as the data in the long- and short-delay conditions, respectively. For DREADD agonist administration, 100 (47 and 53 for monkeys 1 and 2) and 94 (46 and 48 for monkeys 1 and 2) scans were included in the analysis as the data in the long- and short-delay conditions, respectively. To examine the anatomical area in which the fPET activation sites located, we have normalized our $t$-maps of fPET for individual monkeys to publicly available template macaque brain to which a widely used standard macaque atlas[107] has been aligned[108].

ROI analysis was conducted for sessions following hM4Di transduction, with 2-mm radius spheric ROIs around the peak coordinates of activation sites determined in the vehicle administration condition. Mnemonic activation was calculated as the percent activity change in the long-delay condition from the activity at the same ROI obtained in a control, short-delay condition within the same session. To examine the effect of OFC silencing on mnemonic activity at a given ROI, the resultant value of the percent activity change was compared between conditions of vehicle and DREADD agonist administration.

To align the statistical map of mnemonic activation with a CT image of the skull and the penetrated electrode (Fig. 4a and Supplementary Fig. 5a), normalization of fPET images was omitted, and the spatially smoothed fPET images of each monkey were masked with a mask obtained from the structural MR images of the same monkey. Using PMOD, the calculated whole-brain statistical map for each monkey was then coregistered to a CT image of the same monkey obtained when an electrode was penetrated into the region where recorded neurons exhibited a prominent stimulus selective activity during both the cue and delay periods.

## Resting-state fMRI connectivity analysis

Resting-state functional connectivity with the fPET activation sites in the OFC or aVTC as a seed region was calculated using publicly available rs-fMRI data provided by PRIME-DE[45]. We used data of 20 male rhesus macaques acquired by the group of Profs. Mathew Rushworth, Rogier Mars, and Jerome Sallet at the Oxford University under light inhalational anesthesia as described in ref. 44. Briefly, protocols for animal care, MRI, and anesthesia were carried out under authority of personal and project licenses in accordance with the UK Animals (Scientific Procedures) Act (1986) using similar procedures to those that they have previously described[109,110]. During scanning, under veterinary advice, animals were kept under minimum anesthetic using Isoflurane. A four-channel phased-array coil was used (Windmiller Kolster Scientific). The anaesthetized animals were placed in an MRI-compatible stereotaxic frame (Crist Instruments) in a sphinx position and placed in a horizontal 3-T MRI scanner with a full-size bore. Isoflurane was selected for the scans as resting-state networks have been demonstrated previously with this agent[111]. Structural scans were acquired using a T1-weighted MP-RAGE sequence with the following scan parameters: TR, 2,500 ms; TE, 4.01 ms; spatial resolution, $0.5 \times 0.5 \times 0.5$ mm; 128 slices; no spatial gap between slices). Whole-brain BOLD fMRI data in the resting-state were collected for 53 min, 26 s from each animal, via echo-planer imaging (EPI) sequence with the following scan parameters: TR, 2000 ms; TE, 19 ms; 1600 volumes; 36 axial slices; in-plane resolution, $2 \times 2$ mm; slice thickness, 2 mm; no spatial gap between slices.

The above rs-fMRI data were preprocessed using FSL software[100]. The brain masks were created based on the Yerkes19 macaque brain template[99], which was linearly and nonlinearly registered to the individual T1-weighted structural image (T1w) using FMRIB's linear registration tool (FLIRT) and FMRIB's nonlinear registration tool (FNIRT), respectively. Functional images were corrected for motion using motion correction FLIRT (MCFLIRT)[112]. An average functional image was then calculated for each monkey, and was normalized to the T1w using FLIRT and FNIRT. Each functional image was finally normalized to the Yerkes19 macaque template using function-to-T1w and T1w-to-template transformation matrices, followed by spatial smoothing using a Gaussian kernel with the FWHM value of 2.0 mm and band-pass filtering between 0.01 and 0.1 Hz. Seed-to-voxel functional connectivity maps (z-score maps) were calculated for the data obtained from 20 macaques using FMRI Expert Analysis Tool (FEAT) and the volume-of-interests (VOIs) defined as spheres with 2-mm radius around the peak of fPET activation sites (aVTC before hM4Di transduction and OFC in the vehicle condition following hM4Di transduction). Results were plotted on the macaque surface map[106] using Connectome Workbench[105]. Seed-to-ROI connectivity analysis was also conducted as follows. Connectivity between the aVTC node and other nodes that exhibited significant mnemonic activity before hM4Di transduction (OFC, ACC, and IPS) was calculated to derive the network for short-term object memory in which aVTC is embedded. Likewise, connectivity between the OFC node and other nodes showing significant mnemonic activity following hM4Di transduction (aVTC, ACC, IPS, and cEnt) was calculated to compare with the remote effect of OFC silencing.

## Electrophysiological data analysis

Single neuron activities were isolated offline with Offline Sorter (Plexon) based on the analysis of spike waveforms and inter-spike intervals as described previously[64,65,96,113,114]. Neuronal activities during baseline, cue, delay, and choice periods were defined as the average firing rate during the period of 500 ms just before cue onset, 100–300 ms following cue onset (i.e., 100 ms following cue onset to the end of the cue period), 800–3300 ms following cue onset (i.e., 500 ms following cue offset to the end of the delay period), and 100–500 ms following choice onset, respectively. The early, middle,

and late delay periods were defined as 900-ms windows starting from 600, 1500, and 2400 ms following cue offset, respectively. For each neuron, the stimulus selectivity of the response in correct trials was examined using one-way ANOVA, and neurons showing significant ($p < 0.05$) stimulus selectivity during both the cue and delay periods were defined as cue- and delay-selective neurons, and were further analyzed. In total, 50 (29 and 21 from monkeys 1 and 2, respectively) cue- and delay-selective neurons were recorded both before and after DCZ administration in 42 (22 and 20 for monkeys 1 and 2, respectively) recording sessions. Likewise, 28 (14 and 14 from monkeys 1 and 2, respectively) cue- and delay-selective neurons were recorded both before and after vehicle administration in 21 (12 and 9 for monkeys 1 and 2, respectively) recording sessions. Preferred and non-preferred stimuli were determined for each neuron as the stimuli, for which the neuron exhibited the maximum and minimum firing rates during the cue period, respectively. The normalized firing rate was calculated for each neuron as the firing rate normalized to the peak firing rate of the neuron during the cue period for the preferred stimulus. The stimulus selectivity of a given neuron was calculated as the normalized difference between the firing rate for the most preferred stimulus and the mean firing rate for other stimuli[73]. Neurons with high and low stimulus selectivity were determined by median-splitting the cue- and delay-selective neurons into two groups with higher and lower stimulus selectivity. The preservation of the cue stimulus representations during the delay period was assessed for a given neuron by calculating the correlation coefficient between responses during the cue period and the whole or divided sections (i.e., early, middle, and late) of the delay period. The duration of preserved cue representation was calculated for a given neuron as the proportion of sections with significant correlation coefficients. To examine the discriminability between the preferred and non-preferred stimuli, the area under the ROC curve was calculated for each neuron by comparing the distributions of trial-by-trial firing rate for each of these stimuli separately before and during OFC silencing (Supplementary Fig. 6e, f).

To analyze the behavioral relevance of aVTC neuronal activity, a mnemonic error trial was defined as a trial in which monkeys made the wrong choice of the distractor in the choice period of the task, and the firing rate in mnemonic error trials were compared with those in correct trials in the same neurons with the same cue stimulus. In total, the firing rates of 21 and 28 neurons in 23 and 41 mnemonic error trials for the preferred stimulus recorded in 15 and 18 sessions were examined in the control and DCZ administration conditions, respectively.

## Statistics and reproducibility

No statistical method was used to predetermine sample sizes, but they were determined based on the standard in similar studies. Two macaques participated in all the experiments, and the results were consistent between the subjects, suggesting the reproducibility of our results. All attempts for replication were successful. The Investigators were not blinded to allocation during experiments and outcome assessment. No data were excluded from the analyses. Statistical tests were performed as two-tailed unless otherwise stated, and Bonferroni's correction was applied for multiple comparisons.

## Reporting summary

Further information on research design is available in the Nature Portfolio Reporting Summary linked to this article.

# Data availability

All the rs-fMRI data used for the functional connectivity analysis are publicly available at PRIME-DE (https://fcon.1000.projects.nitrc.org/indi/indiPRIME.html). The template macaque brain to which a widely used standard macaque atlas[107] has been aligned is publicly available at https://www.bic.mni.mcgill.ca/ServicesAtlases/Macaque. The data generated in this study have been deposited in GitHub (https://github.

com/CAToshiyuki/DMS-2024). Source data are provided with this paper.

## Code availability

All the in-house codes used in the present study have been deposited in GitHub (https://github.com/CAToshiyuki/DMS-2024).

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

## Acknowledgements

We thank R. Suma, T. Kokufuta, R. Yoshida, T. Sugii, R. Yamaguchi, Y. Matsuda, J. Kamei, M. Nakano, and M. Fujiwara for their technical assistance, and Dr. S. Bullet for helpful discussion. We also thank to K. Kumata and Dr. M.R. Zhang at Department of Radiopharmaceuticals Development, QST for producing the radioligands. This work was supported by AMED Grant JP23dm0307007 (to T.H.), JP20dm0307021 (K.I.), JP21dm0207077 (M.T.), JP20dm0107146 (T.M.), MEXT/JSPS KAKENHI JP17H02219 and JP24H00734 (T.H.), JP22H05157 (K.I.), JP19H05467 (M.T.), JP15H05917 and JP20H05955 (T.M.), The commissioned research by the NICT Grant 22301 (T.M.), National Bio-Resource Project "Japanese Monkeys" of MEXT.

## Author contributions

Conceptualization: T.H. Methodology: T.H., Y.N., Yuki Hori, Yukiko Hori, N.M., and K.I. Investigation: T.H., Yuki Hori, and Yukiko Hori. Visualization: T.H. and T.M. Funding acquisition: T.H., K.I., M.T., and T.M. Project administration: T.H. and T.M. Supervision: T.H. and T.M. Writing—original draft: T.H. Writing—review & editing: T.H., Y.N., Yuki Hori, Yukiko Hori, K.O., K.M., N.M., H.I., K.I., T.S., M.T., M.H., and T.M.

## Competing interests

The authors declare no competing interests.
