## [Peer Review File · Nature Communications]

REVIEWER COMMENTS

Reviewer #1 (Remarks to the Author):

In their paper, Hirabayashi present a truly remarkable picture of functional interactions between targeted regions of orbitofrontal cortex and the anterior temporal cortex. By combining task relevant PET imaging, analysis of resting state fMRI data, and chemogenetic silencing in the context of a memory task with variable delays, they show convincingly that OFC exerts critical top down modulation of activity in temporal cortex. This paper is a major advance not only because of the incredible set of tools brought to bear on this outstanding question, but also because their data driven approach led them to target regions of OFC as opposed to other regions of the PFC that the current state of the literature might have suggested (in particular DLPFC). This paper illustrates the power of reproducible DREADD based silencing and will surely have significant impact on the cognitive neuroscience community.

General comments

In general, the paper is extremely well written and very clear. The careful attention to detail and consistent reference to control conditions indicating that their results were not artifactual amplify the power of this report.

1. Discussion of LPFC. I found the discussion of LPFC (pp. 20 and following) to be excellent (I was wondering if/when they would address this issue). I'm a bit torn, but wonder if mentioning this in the Introduction might be useful just to make it clear that the authors will discuss this in detail.

2. Novel objects. This is an important point, and the authors again discuss that this study focuses on familiar objects. I wonder if the authors might have any thoughts not only about changes in responses in both OFC and aVTC but whether the top down modulation from OFC to aVTC requires familiarity?

3. Delays tested. This was not entirely clear to me. Is it correct that during PET scanning long delays were 3 or 4 secs but during DREADD inactivation, delays up to 9 secs were used (Figure 2)?

4. Eye movements during the behavioral task. During the physiology testing, which seems quite convincing, I was not sure how the monkeys' eye movements during the decision phase affecting

the overall activity? Presumably they can look at each object before touching the screen? This would likely affect the recordings in, e.g. Figure 5C?

5. Correct vs error trials. I think the attempt to divide trials between correct and incorrect trials is interesting, but I'm not sure that it's clear from Figure 6E that the difference in the DSZ conditions goes away because the performance during the delay is so affected that there is not room for a difference to be observed? If the performance dropped to around 75% overall, then many of the "correct" trials are likely a result of guessing, so it's not clear that these would be expected to be so different in terms of normalized delta activity in error trials. (The data shown in Figure S6 also aren't so easy to interpret.)

Minor comments

p. 28 "retracing" should be "retracting" I believe

p. 35 "basically" seems like awkward usage. I think this might mean "Baseline neuronal activity was recorded...".

p. 63 Figure 6C and 6F I'm not sure what the word "Devison" is?

Supplemental Figure S2: Not sure why the DCZ in Monkey 2 is shown there. Is this different from the data in Figure 2A?

Supplemental Figure S4E: The change in success rate seems less than 5% - is this collapsed across delays (it seems small compared to changes in performance shown elsewhere).

Reviewer #2 (Remarks to the Author):

In this manuscript, Hirabayashi et al examined the effects of orbital frontal inactivation on memory guided behavior. They did use a variety of neuroimaging, electrophysiological and behavioral methods to investigate how chemogenetic silencing of the OFC changes nonhuman primates' behavior on a classic delayed memory task. They found that chemogenetic silencing of the OFC had a profound effect on the animal's behavior, particularly at longer memory delays. Additionally,

they found that this performance change was associated with changes in activity at the both the systems (PET) and single unit level in the anterior temporal lobe and OFC.

Overall, the manuscript presents an impressive set of studies and analyses that span a broad range of techniques. The combination of simultaneous PET and behavior, electrophysiology and behavior, as well as resting-state fMRI and chemogenetics is unique and provides a well rounded and robust investigation into the underlying interaction between the OFC and memory related behavior. Despite these positives, there are a number of concerns I have based on some of data presentation and statistical methods employed, that I feel need further addressing.

With some further details and possibly analyses, I believe this manuscript will have a significant impact on the field.

Listed below are some of the concerns and questions I have for the current version of the manuscript.

Major Points:

The authors claim that the OFC and aVTC activity is consistent across their two animals. Which leads them to using group data from PRIME-DE to calculate resting-state networks based on their ROIs. While the hotspots are nearby in figure S1B, it clear that there are differences in location. It would be useful to understand how far apart the regions are and their atlas-based location. In particular, it looks like the OFC activations could potentially be in different regions of the OFC as they appear to be a few mm apart. Displaying the subjects on the same image after warping to standard space could help, or warping an atlas to the subjects' space and showing that the activity is in the same region could be useful. Also using the PRIME-DE data it should be possible to calculate resting-state network profiles for ROI that specifically overlap the individual subject's memory activity. These analyses would help to shed light on how similar the activation is across subjects. This issue is of particular relevance when it comes to the DREADDs inhibition compared to the vehicle condition. The authors claim that similar areas were active during vehicle and baseline but the data presented in Figures S1 and S3 seem to paint a different picture where within animal different regions are being found. Additionally, on page 9, lines 22-24, the authors claim that the IPS was active in both the control and inhibition condition, but the activity shown is in opposite hemispheres. All together the lack of direct comparisons between regions and playing a bit loose with homologies makes it difficult to interpret the actual results.

There seems to be some confusion within the manuscript about where the effects of CNO versus DCZ are being displayed. Page 8 suggest that the CNO data should be either in Figure 2A or Figure

S2B. Figure 2A is not specific about what agonist was used. If I'm reading the results correctly in Figure 2A M1 is displaying DCZ data, whereas M2 is showing CNO, with M2s DCZ data being shown in Figure S2B. This organization seems to be done to possibly mask a small difference in agonist within M2. The authors state in lines 4-8 that there is no difference between agonists, but don't present a statistic to back that up, and instead seem to show an across subject and agonist comparison where a within subject across agonist comparison would be far more appropriate.

Page 9 line 18, why is the histological data not shown?

The result section is lacking of necessary details about the neural recordings, such as the number of sessions, and number neurons. Number of trials recorded in each condition. These descriptors help to inform the reader about how reliable any result being show may be. How consistent were the delay period activity? That is did all neurons show this effect or was it only a subset, the majority of the statistics being used are population level effects thus not tell the reader much about the specificity of those changes to different neurons.

Minor Points:

Page 9 line 16, re-fMRI should probably be rs-fMRI.

Our point-by-point response to the reviewers' comments are written in blue.

Reviewer #1:

In their paper, Hirabayashi present a truly remarkable picture of functional interactions between targeted regions of orbitofrontal cortex and the anterior temporal cortex. By combining task relevant PET imaging, analysis of resting state fMRI data, and chemogenetic silencing in the context of a memory task with variable delays, they show convincingly that OFC exerts critical top down modulation of activity in temporal cortex. This paper is a major advance not only because of the incredible set of tools brought to bear on this outstanding question, but also because their data driven approach led them to target regions of OFC as opposed to other regions of the PFC that the current state of the literature might have suggested (in particular DLPFC). This paper illustrates the power of reproducible DREADD based silencing and will surely have significant impact on the cognitive neuroscience community.

We appreciate your positive evaluation of our study.

General comments

In general, the paper is extremely well written and very clear. The careful attention to detail and consistent reference to control conditions indicating that their results were not artifactual amplify the power of this report.

Again, thank you for your positive evaluation of our study.

1. Discussion of LPFC. I found the discussion of LPFC (pp. 20 and following) to be excellent (I was wondering if/when they would address this issue). I'm a bit torn, but wonder if mentioning this in the Introduction might be useful just to make it clear that the authors will discuss this in detail.

We appreciate your suggestion. In accordance with this comment, in the Introduction of the revised manuscript, we have mentioned that the issue of possible LPFC involvement will be discussed later, as follows: "...we address the issue of the possible involvement of other brain regions, such as the lateral prefrontal cortex (LPFC), in the Discussion..."

2. Novel objects. This is an important point, and the authors again discuss that this study focuses on familiar objects. I wonder if the authors might have any thoughts not only about changes in responses in both OFC and aVTC but whether the top down modulation from OFC to aVTC requires familiarity?

Thank you for your comment. We suspect that if the responses in the OFC or aVTC depended on familiarity, then the top down modulation between them would also depend on familiarity. Note that this does not necessarily mean that the top down modulation requires familiarity, but rather that the top down modulation in the case of novel stimuli would likely be somewhat different from that observed in the present investigation. However, a clearer answer to this question requires experimental confirmation in future studies. We have added a description of this concept to the Discussion of the revised manuscript.

3. Delays tested. This was not entirely clear to me. Is it correct that during PET scanning long delays were 3 or 4 secs but during DREADD inactivation, delays up to 9 secs were used (Figure 2)?

During behavioral testing without physiological measurements, where we closely examined the behavioral effects of DREADD inactivation, delays of up to 9 s were used (Figure 2). During PET scanning, however, long delays were 3 or 4 secs for both the DREADD inactivation and non-inactivation conditions, as a compromise between sufficient mnemonic demand and sufficient trials during the scanning period (1 minute). We would like to emphasize that even in this condition, significant behavioral effects were observed during the PET scanning (as shown in Figure S3G in the revised manuscript).

4. Eye movements during the behavioral task. During the physiology testing, which seems quite convincing, I was not sure how the monkeys' eye movements during the decision phase affecting the overall activity? Presumably they can look at each object before touching the screen? This would likely affect the recordings in, e.g. Figure 5C?

The monkeys can look at each object before touching the screen, as you have pointed out; however, they only did so after the onset of the choice period, during which the objects to touch were presented. Figure 5C shows the neuronal activity *before* the onset of the presentation of objects to touch, when the monkeys' eye movements had not yet begun to search for the target object. We also analyzed the neuronal activity during the

choice period, but only in one Supplemental figure panel (Figure S5L in the revised manuscript). We note that although the neuronal activity during the choice period might have been affected by eye movements, the effect of OFC silencing was not observed during this period. In the revised manuscript, we have described these points in the legends of Figures 5C and S5L.

5. Correct vs error trials. I think the attempt to divide trials between correct and incorrect trials is interesting, but I'm not sure that it's clear from Figure 6E that the difference in the DSZ conditions goes away because the performance during the delay is so affected that there is not room for a difference to be observed? If the performance dropped to around 75% overall, then many of the "correct" trials are likely a result of guessing, so it's not clear that these would be expected to be so different in terms of normalized delta activity in error trials. (The data shown in Figure S6 also aren't so easy to interpret.)

We appreciate your insights into this matter. Certainly, if the performance dropped to around 75% overall, then many of the "correct" trials might simply be the result of guessing; it would then be unclear whether the normalized activity was expected to be so different between the correct and error trials—or there might be no room for a difference to be observed. However, the electrophysiological experiments were conducted with a 3-sec delay condition. In this condition, the correct rate was above 90% on average even after OFC silencing, and the average difference in performance between the conditions with and without silencing was less than 5% (although this difference was statistically significant, as shown in Figure S5E in the revised manuscript). We have added this description to the Results in the revised manuscript.

Regarding the data shown in Figure S6 in the original manuscript (Figure S7 in the revised manuscript), panel A depicts scatter plots comparing the normalized firing rate of individual neurons in correct and error trials for cue and delay periods. These plots show that activity in the error trials decreased during the delay, but not the cue, period in each neuron. To compare the error-predicting activity decrease during these periods more directly, we plotted the decreases in firing rates in error trials during the cue and delay periods on the x and y axes, respectively, in panel B. If the activity decrease in the error trials during the cue period was caused by the monkeys' failure to watch the cue object correctly, and this resulted in the activity decrease during the subsequent delay period, then the changes in firing rates during these two periods would be expected to correlate with each other. However, no such correlation was identified, suggesting that the observed decrease in delay activity was unlikely to reflect such a

perceptual failure. Panel C shows the data same as in A, but under OFC silencing conditions. Please note that we have added some of the above descriptions to the Figure S6 legend in the revised manuscript, to improve the clarity of this figure.

Minor comments

p. 28 "retracing" should be "retracting" I believe

We have corrected this. Thank you.

p. 35 "basically" seems like awkward usage. I think this might mean "Baseline neuronal activity was recorded...?"

Thank you for pointing this out. Here, we wanted to note the smallest number of trials for each object. However, this number was not strictly set, but rather fluctuated session by session. We thus described it as "...recorded basically for at least 10 trials for each stimulus" in the original manuscript. To improve the clarity of this text, we have revised the sentence to read: "Neuronal activity was recorded for approximately 10 or more trials for each stimulus..."

p. 63 Figure 6C abd 6F I'm not sure what the word "Devision" is?

We have corrected the word "Devision" to "Division" in Figures 6C, 6F, and 5F. Thank you for pointing this out.

Supplemental Figure S2: Not sure why the DCZ in Monkey 2 is shown there. Is this different from the data in Figure 2A?

The data from monkey 2 in Figure 2A was showing the effects of another DREADD agonist, CNO. To clarify this point, we have added descriptions of the specific agonist used for each monkey to the Figure 2A legend and the Results in the revised manuscript. In association with the data from monkey 2 in Figure 2A, Supplemental Figure S2 shows that both DREADD agonists had a similar effect, which suggests that the effect was not dependent on the peripheral properties of a specific reagent. Moreover, in response to a related comment from Reviewer 2, we have statistically confirmed that the differences in effects between CNO and DCZ were not significantly different (see our response to a Reviewer #2's comment for details).

Supplemental Figure S4E: The change in success rate seems less than 5% - is this collapsed across delays (it seems small compared to changes in performance shown elsewhere).

The change in success rate in Supplemental Figure S4E in the original manuscript (Figure S5E in the revised manuscript) was for a 3-s delay, which was roughly equivalent to the corresponding data (i.e., 3-s delay) in Figures 2A and S2B, but much smaller than that for a 9-s delay in these figure panels. We have described this point in the Figure S5E legend in the revised manuscript.

We appreciate the Reviewer for all of his/her helpful suggestions throughout.

Reviewer #2 (Remarks to the Author):

In this manuscript, Hirabayashi et al examined the effects of orbital frontal inactivation on memory guided behavior. They did used a variety of neuroimaging, electrophysiological and behavioral methods to investigate how chemogenetic silencing of the OFC changes nonhuman primates' behavior on a classic delayed memory task. They found that chemogenetic silencing of the OFC had a profound effect on the animal's behavior, particularly at longer memory delays. Additionally, they found that this performance change was associated with changes in activity at the both the systems (PET) and single unit level in the anterior temporal lobe and OFC.

Overall, the manuscript presents an impressive set of studies and analyses that span a broad range of techniques. The combination of simultaneous PET and behavior, electrophysiology and behavior, as well as resting-state fMRI and chemogenetics is unique and provides a well rounded and robust investigation into the underlying interaction between the OFC and memory related behavior.

We appreciate your positive evaluation of our study.

Despite these positives, there are a number of concerns I have based on some of data presentation and statistical methods employed, that I feel need further addressing. With some further details and possibly analyses, I believe this manuscript will have a significant impact on the field.

Listed below are some of the concerns and questions I have for the current version of the manuscript.

We appreciate your constructive comments. Indeed, we believe that addressing these comments—as you can see below in a point-by-point manner—has strengthened our study.

Major Points:

The authors claim that the OFC and aVTC activity is consistent across their two animals. Which leads them to using group data from PRIME-DE to calculate resting-state networks based on their ROIs. While the hotspots are nearby in figure S1B, it clear that

there are differences in location. It would be useful to understand how far apart the regions are and their atlas-based location. In particular, it looks like the OFC activations could potentially be in different regions of the OFC as they appear to be a few mm apart. Displaying the subjects on the same image after warping to standard space could help, or warping an atlas to the subjects' space and showing that the activity is in the same region could be useful.

We appreciate your constructive suggestions, and acknowledge the importance of accurately determining the anatomical locations of OFC and aVTC activities. In response to your comments, we have taken additional steps to accurately localize the activations within a standard anatomical framework.

In the original manuscript, we indeed displayed the subjects on the same image after warping to standard space. In the revised manuscript, we have further analyzed the spatial locations of the activations by normalizing our *t*-maps of fPET for each individual monkey to a publicly available macaque brain template to which a widely used standard macaque atlas (Paxinos et al., 2008) has been aligned (Frey et al., 2011). According to the aforementioned atlas, the activation peaks of both monkeys shown in Figure S1 were located in the same anatomical area (areas 13 and TE1 for the OFC and aVTC, respectively). We also quantified the distances between the activation peaks in both monkeys and found that the distances were 6.3 and 5.5 mm for the OFC and aVTC, respectively. These measurements confirmed that, although minor spatial differences were observed, the activity was in the same region across subjects. We have added these descriptions to the Figure S1 legend in the revised manuscript.

Also using the PRIME-DE data it should be possible to calculate resting-state network profiles for ROI that specifically overlap the individual subject's memory activity. These analyses would help to shed light on how similar the activation is across subjects. This issue is of particular relevance when it comes to the DREADDs inhibition compared to the vehicle condition. The authors claim that similar areas were active during vehicle and baseline but the data presented in Figures S1 and S3 seem to paint a different picture where within animal different regions are being found.

We appreciate your insightful suggestion to further examine the functional connectivity profiles using rs-fMRI data, to assess the consistency of activation sites across subjects and conditions. In response to your comments, we have added analyses focusing on the resting-state network profiles for ROIs that specifically overlap with each individual

subject's memory activity under both baseline (i.e., *before* DREADD expression) and vehicle (i.e., *after* DREADD expression) conditions. The results are presented in a new figure panel (Figure S3D in the revised manuscript).

In these new analyses, we found that when seeded from the OFC activation sites in each of the before and after DREADD expression conditions in each monkey, significant connectivity was observed at the same activation site in the aVTC that was identified in the group analysis after DREADD expression ($P < 0.001$ for both conditions in both monkeys). Likewise, when seeded from the aVTC activation sites in each condition in each monkey, significant connectivity was observed at the same OFC activation site that was identified in the group analysis ($P < 0.001$ and 0.002 for monkeys 1 and 2, respectively). These results suggest that the activation sites before and after DREADD expression were similar to each other, at least in terms of the fronto-temporal rs-fMRI connectivity that was the particular focus of the study.

Furthermore, to examine the overall consistency between the connectivity values in the two conditions, we calculated the connectivity between four OFC or aVTC activation sites (i.e., the baseline and vehicle conditions for each monkey) and the other activation sites that were identified in the group analysis following DREADD expression (i.e., the aVTC, OFC, ACC, IPS, and cEnt as shown in Figures 3A and S4A in the revised manuscript). As a whole, these connectivity values were significantly correlated between the conditions ($P < 0.003$), and this significant correlation was maintained for each monkey ($P < 0.003$ and 0.001 for monkeys 1 and 2, respectively). In the revised manuscript, we have added a figure panel showing this correlation (Figure S3E). These additional analyses support our claim that similar areas within the OFC and aVTC were active in both monkeys during the baseline condition (*before* DREADD expression) and the vehicle condition (*after* DREADD expression), at least in terms of rs-fMRI connectivity along the network associated with short-term visual object memory. In the revised manuscript, we have added these descriptions to the legends of Figures S3D and S3E.

Additionally, on page 9, lines 22-24, the authors claim that the IPS was active in both the control and inhibition condition, but the activity shown is in opposite hemispheres. All together the lack of direct comparisons between regions and playing a bit loose with homologies makes it difficult to interpret the actual results.

To validate the homology between the IPS activations in both hemispheres, we overlaid a macaque cortical atlas onto our normalized activation maps to locate the IPS activations within an anatomical framework. We then compared the y and z coordinates

of the activation peaks. After adjusting for hemispheric differences by flipping the activation site in the DCZ condition from the right to the left hemisphere, the distance between the activation peaks was 2.2 mm.

In the revised manuscript, we have therefore included a new figure (Figure S4A, middle bottom, green) that visually represents the superimposed activation sites in the vehicle and DCZ conditions within the same slice. This figure demonstrates the proximity of the IPS activation sites in the two conditions, supporting the idea that those activation peaks were located within the same anatomical region, which was specifically identified as area PEa according to Paxinos et al., 2008. Notably, previous research indicated that area PEa is associated with retrieval success signal in a recognition memory task (Miyamoto et al., 2013).

These results, which we have detailed in our revised manuscript, suggest that the observed IPS activation sites in the vehicle and DCZ conditions were indeed located in the homologous brain region involved in relevant cognitive function, despite occurring in opposite hemispheres. We again thank you for your comments here, which helped to strengthen our results.

There seems to be some confusion within the manuscript about where the effects of CNO versus DCZ are being displayed. Page 8 suggest that the CNO data should be either in Figure 2A or Figure S2B. Figure 2A is not specific about what agonist was used. If I'm reading the results correctly in Figure 2A M1 is displaying DCZ data, whereas M2 is showing CNO, with M2s DCZ data being shown in Figure S2B. This organization seems to be done to possibly mask a small difference in agonist within M2. The authors state in lines 4-8 that there is no difference between agonists, but don't present a statistic to back that up, and instead seem to show an across subject and agonist comparison where a within subject across agonist comparison would be far more appropriate.

In the Results and the Figure 2A legend in the revised manuscript, we have added a description stating that DCZ and CNO were used for monkeys 1 and 2, respectively, to clarify which DREADD agonists were used for the behavioral testing. Moreover, to statistically analyze the differences between the effects of CNO and DCZ within one animal (monkey 2), we performed a three-way ANOVA with the following factors: Agonist (i.e., CNO or DCZ), Silencing (i.e., vehicle or DREADD agonist), and Delay length. Although the interaction between the factors of Silencing and Delay length remained significant ($P < 0.001$), there was no significant interaction across all the three factors ($P > 0.1$), indicating that both agonists had similar behavioral effects as a function of the

delay length, and that there were no significant differences in the effects of these agonists. In the revised manuscript, we have added these descriptions to the Results and the Figure S2 legend.

Page 9 line 18, why is the histological data not shown?

We appreciate your interest regarding the lack of histological data in the original manuscript. In response to your question and to provide further evidence for our findings, we have included histological data in the revised manuscript. Specifically, we have added Figure S4D, which shows histological data demonstrating that GFP-expressing axonal fibers of OFC neurons were detected in the aVTC. These data support our claim that DREADD-expressing OFC neurons directly connect to the aVTC, which is consistent with the findings of previous anatomical studies (Mohedano-Moriano *et al.* 2015; Saleem *et al.*, 2008; Giarrocco & Averbeck 2021). In the revised manuscript, we have added a description to this effect in the Figure S4D legend.

The result section is lacking of necessary details about the neural recordings, such as the number of sessions, and number neurons. Number of trials recorded in each condition. These descriptors help to inform the reader about how reliable any result being show may be.

In the revised manuscript, we have added necessary details about the neural recordings to the Results, such as the number of sessions, the number of neurons, and the average number of trials recorded in each condition. Overall, 49 cue- and delay-selective neurons were recorded both before and after DCZ administration in 42 recording sessions. The numbers of trials recorded before and after DCZ administration were 10.9 ± 0.4 and 11.3 ± 0.5 (mean \pm sem), respectively.

How consistent were the delay period activity? That is did all neurons show this effect or was it only a subset, the majority of the statistics being used are population level effects thus not tell the reader much about the specificity of those changes to different neurons.

We have examined the statistical significance of the effects of OFC silencing on delay activity for individual neurons. Of the total population of neurons that was analyzed (i.e., delay selective neurons recorded both before and after OFC silencing), 51% (25 of 49 neurons) exhibited a statistically significant reduction in the delay period activity for the

preferred stimulus at the single-neuron level ($P < 0.05$, unpaired t -test). No single neuron showed the opposite effect (i.e., significant *enhancement* of the delay activity for the preferred stimulus). We have added these descriptions to the Results in the revised manuscript. We appreciate your insightful comment, which led to the strengthening of our electrophysiological data.

Minor Points:

Page 9 line 16, re-fMRI should probably be rs-fMRI.

We have corrected this error. Thank you.

We appreciate the reviewer for all of his/her helpful suggestions throughout.

References

Frey, S., Pandya, D.N., Chakravarty, M.M., Bailey, L., Petrides, M. & Collins, D.L. An MRI based average macaque monkey stereotaxic atlas and space (MNI monkey space). *Neuroimage* **55**, 1435-1442. doi: 10.1016/j.neuroimage.2011.01.040 (2011).

Giarrocco, F. & Averbeck, B. B. Organization of parietoprefrontal and temporoprefrontal networks in the macaque. *J Neurophysiol* **126**, 1289-1309 (2021).

Miyamoto, K. *et al.* Functional differentiation of memory retrieval network in macaque posterior parietal cortex. *Neuron* **77**, 787-799, doi:10.1016/j.neuron.2012.12.019 (2013).

Mohedano-Moriano, A. *et al.* Prefrontal cortex afferents to the anterior temporal lobe in the *Macaca fascicularis* monkey. *J Comp Neurol* **523**, 2570-2598 (2015).

Paxinos, G., Huang, X. -F., Petrides, M., & Toga, A. W. The Rhesus Monkey Brain in Stereotaxic Coordinates. Academic Press, San Diego. (2008).

Saleem, K. S., Kondo, H. & Price, J. L. Complementary circuits connecting the orbital

and medial prefrontal networks with the temporal, insular, and opercular cortex in the macaque monkey. *J Comp Neurol* **506**, 659-693 (2008).

REVIEWERS' COMMENTS

Reviewer #1 (Remarks to the Author):

The revised manuscripts addresses all of my concerns. Again, the authors multimodal approach to this highly relevant experimental question - relating top down control to sensory memory, is extremely interesting. The role of OFC, which has not been the focus of many previous studies, is particularly striking and will surely have significant impact.

Reviewer #2 (Remarks to the Author):

The authors have done an excellent job of responding to the previous round of comments. Their updated manuscript more clearly states their results and helps the reader to understand the data and implications of them. On the whole, I feel that the revised manuscript is considerably more polished and I have no further specific concerns at this time.